# Endocrine remodelling of the adult intestine sustains reproduction in *Drosophila*

**Tobias Reiff[1†], Jake Jacobson[2†], Paola Cognigni[2†‡], Zeus Antonello[1†], Esther Ballesta[1], Kah Junn Tan[3], Joanne Y Yew[3,4§], Maria Dominguez[1]\*, Irene Miguel-Aliaga[2]\***

[1]Instituto de Neurociencias, Consejo Superior de Investigaciones Científicas, Universidad Miguel Hernández, Alicante, Spain; [2]MRC Clinical Sciences Centre, Imperial College London, London, United Kingdom; [3]Temasek Life Sciences Laboratory, Singapore, Singapore; [4]Department of Biological Sciences, National University of Singapore, Singapore, Singapore

**Abstract** The production of offspring is energetically costly and relies on incompletely understood mechanisms that generate a positive energy balance. In mothers of many species, changes in key energy-associated internal organs are common yet poorly characterised functionally and mechanistically. In this study, we show that, in adult *Drosophila* females, the midgut is dramatically remodelled to enhance reproductive output. In contrast to extant models, organ remodelling does not occur in response to increased nutrient intake and/or offspring demands, but rather precedes them. With spatially and temporally directed manipulations, we identify juvenile hormone (JH) as an anticipatory endocrine signal released after mating. Acting through intestinal bHLH-PAS domain proteins Methoprene-tolerant (Met) and Germ cell-expressed (Gce), JH signals directly to intestinal progenitors to yield a larger organ, and adjusts gene expression and sterol regulatory element-binding protein (SREBP) activity in enterocytes to support increased lipid metabolism. Our findings identify a metabolically significant paradigm of adult somatic organ remodelling linking hormonal signals, epithelial plasticity, and reproductive output.

**\*For correspondence:**
m.dominguez@umh.es (MD);
i.miguel-aliaga@imperial.ac.uk (IM)

[†]These authors contributed equally to this work

**Present address:** [‡]Centre for Neural Circuits and Behaviour, University of Oxford, Oxford, United Kingdom; [§]Pacific Biosciences Research Center, University of Hawai'i at Mānoa, Honolulu, United States

**Competing interests:** The authors declare that no competing interests exist.

## Introduction

Reproduction is energetically costly. Mothers can adjust their energy balance to maximise their reproductive success through well-established neural mechanisms that match food intake to their enhanced energy requirements (*Roa and Tena-Sempere, 2014*). However, less well-understood changes also occur in many animals during reproduction; internal organs, such as the liver, pancreas, and gastrointestinal tract, increase their size and adapt their physiology, potentially contributing to an increased generation and delivery of nutrients (*Hammond, 1997*; *Speakman, 2008*).

Establishment of a positive energy balance may be particularly important to animals with a reproductive strategy that involves rapid production of large numbers of progeny. *Drosophila melanogaster* females can lay up to 100 eggs per day at the peak of their fertility in early life (*David et al., 1974*; *Klepsatel et al., 2013*). We hypothesised that such demands may rely on major regulatory responses, which are amenable to genetic investigation in this model system. A network of organs and tissues in *Drosophila* perform many of the same basic functions as those found in mammals (*Padmanabha and Baker, 2014*), so we sought to explore the nature and significance of organ plasticity during reproduction.

**eLife digest** Producing offspring places extra energy demands on individuals. Female animals—which generally invest more time and resources than the males—need to ensure that sufficient nutrients reach their offspring during pregnancy and lactation. The small intestines of many female animals increase in size during this period, but it is not clear to what extent these changes help to maximise reproduction, or how they are regulated.

Reiff, Jacobson, Cognigni, Antonello et al. investigated what happens to the middle section of the gut in female fruit flies after mating. A fly's 'midgut' performs a similar role to the small intestine in humans and other mammals. The experiments show that mating increases the numbers of cells in the midgut so that it increases in size.

These changes are due to a hormone called 'juvenile hormone', which is released after the fly mates. In particular cells of the midgut, juvenile hormone also regulates some genes involved in the metabolism of lipids. If the activity of juvenile hormone is blocked in these cells, the female flies produce fewer eggs. These changes in the midgut still happen in mutant flies that cannot produce eggs and don't increase their food intake after they mate. Therefore, the changes appear to prepare flies for the increased nutritional demands rather than being caused by it.

Altogether, these findings reveal that changes in the midgut play an important role in the ability of female fruit flies to reproduce. Similar changes to the gut may also increase reproductive success in humans and other mammals. However, if the changes are maintained after reproduction, it is possible that they may contribute to weight gain and an increased risk of cancer in females after pregnancy.

## Results

### The adult midgut is remodelled in female flies after mating

Female flies undergo multiple post-mating adaptations including changes in digestive physiology (*Cognigni et al., 2011*). This prompted us to characterise possible intestinal changes occurring during the phase of peak fertility (*David et al., 1974*; *Klepsatel et al., 2013*). We focused on the midgut epithelium because of its major digestive/absorptive roles (*Lemaitre and Miguel-Aliaga, 2013*). In the midgut epithelium, long-lived progenitors (intestinal stem cells (ISCs)) divide to self-renew and to give rise to committed progenitors (called enteroblasts (EBs)), which directly differentiate into two types of progeny: absorptive enterocytes (ECs) and enteroendocrine cells (EECs) (*Jiang and Edgar, 2012*). We found that mating increases the number of both dividing and differentiating midgut cells, as revealed by phospho-Histone H3 (pH3) stainings and temporal analyses of progenitors and their descendants using the dual-labelling system *escargot*-Repressible Dual Differential Marker (*esgReDDM*, *Antonello et al., 2015*) (*Figure 1A,C,E*). The midgut of mated females also becomes visibly larger; gut diameter measurements were suggestive of a net increase in the number of postmitotic intestinal cells (*Figure 1B,D*, *Figure 1—figure supplement 1*): an increase that we confirmed by cell number and density counts (*Figure 1F*, *Figure 1—figure supplement 1*). Concurrent with midgut re-sizing, we observed mating-dependent activation of the single *Drosophila* homologue of the mammalian family of sterol regulatory element-binding proteins (SREBPs [*Theopold et al., 1996*; *Shimano, 2001*; *Seegmiller et al., 2002*], also known as *HLH106* in flies, *Figure 2A,B*), using a reporter subject to the same physiologically regulated proteolytic processing as wild-type SREBP (*Kunte et al., 2006*). SREBP activation after mating was accompanied by upregulation of midgut transcripts involved in fatty acid synthesis and activation (*SREBP*, the long-chain fatty acid CoA ligases *bubblegum* (*bgm*) and *Acyl-CoA synthetase long-chain* (*Acsl*) and, depending on genetic background, *Fatty acid synthase* (*FAS*) and *Acetyl-CoA carboxylase* (*ACC*)) (*Figure 2E,F*), many of which are known SREBP targets in flies and/or mammals (*Seegmiller et al., 2002*; *Horton et al., 2003*). Immunohistochemical analyses using reporters pointed to the ECs located in the posterior midgut region (R5, *Buchon et al., 2013*; *Marianes and Spradling, 2013*) as preferential sites of transcriptional and SREBP activity changes (*Figure 2A–D*). Thus, in female flies actively engaged in reproduction, changes in both intestinal progenitors and their progeny parallel those observed in mammals leading to hyperplasia

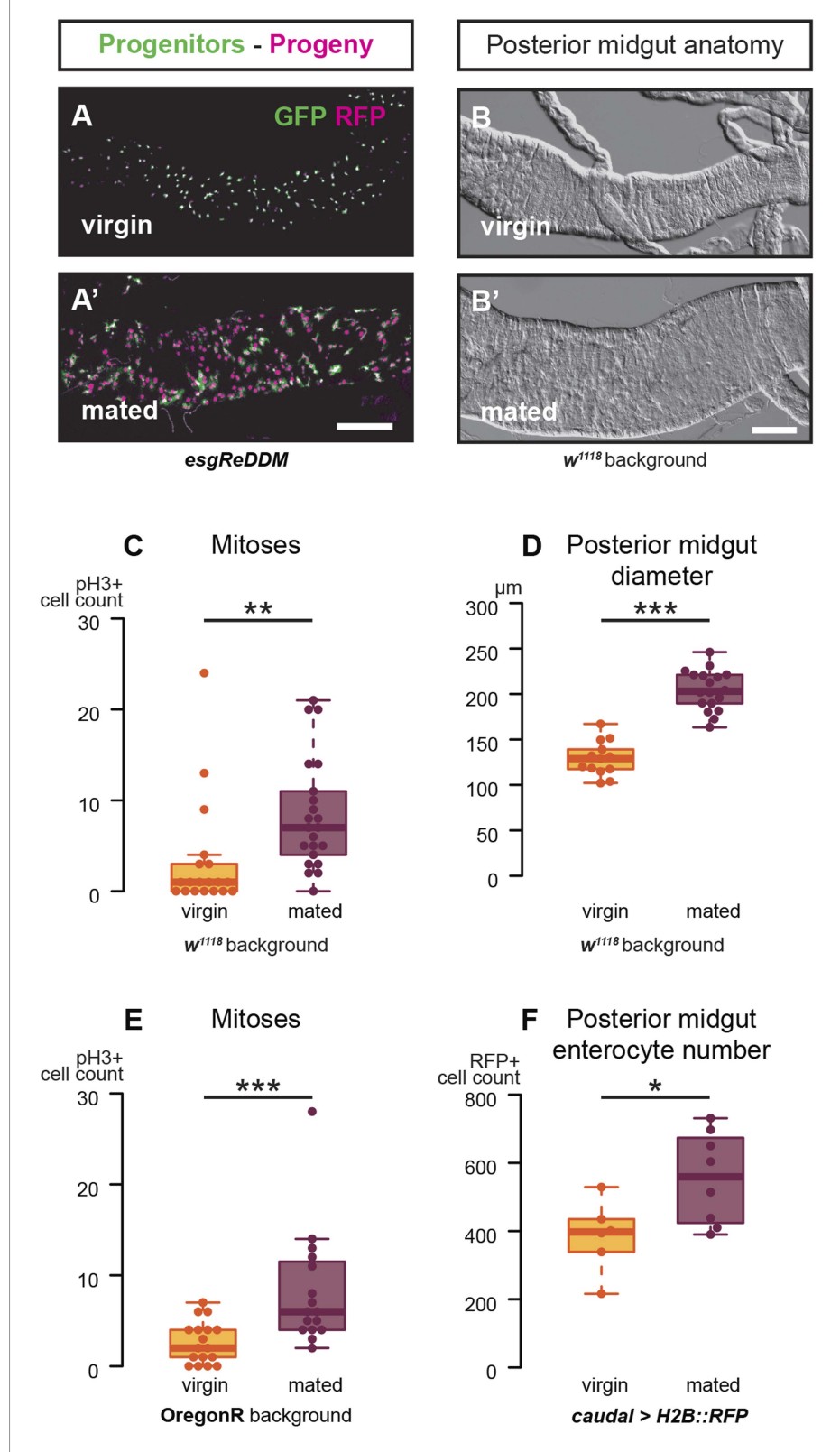

**Figure 1**. Mating increases ISC proliferation and gut size. (**A**, **A'**) Using the *esgReDDM* tracer (*Antonello et al., 2015*), intestinal progenitors (intestinal stem cells (ISCs) and enteroblasts) are labelled with GFP and RFP, whereas the postmitotic progeny (enterocytes (ECs) and enteroendocrine cells) that these progenitors give rise to in
*Figure 1. continued on next page*

*Figure 1. Continued*

a defined time window is labelled with RFP only (see Supplemental Information for additional details). At 3 days after mating, the posterior midgut of mated flies contains more newly generated postmitotic progeny (**A**) compared to age-matched virgins (**A′**). It has also become visibly larger (**B**, **B′**). At this time point, these guts also have a higher number of nuclei marked by the mitotic marker pH3 in both $w^{1118}$ and OregonR backgrounds (**C**, $p = 0.008$, and **E**, $p < 0.001$, negative binomial GLM), although the proliferation increase is transient (data not shown). The size increase is quantified in the posterior midgut by measuring midgut diameter (**D**, $p < 0.001$, t-test) and counting the number of cells labelled by the EC marker *caudal-Gal4* (**F**, $p = 0.02$, t-test). See *Table 1* for full genotypes.

The following figure supplement is available for figure 1:

**Figure supplement 1**. Mating re-sizes the *Drosophila* gut.

(*Hammond, 1997*; *Speakman, 2008*), increased organ size (*Hammond, 1997*; *Speakman, 2008*) and upregulation of lipid gene expression (*Athippozhy et al., 2011*).

## Intestinal remodelling is mediated by increased levels of circulating juvenile hormone

Female flies change their physiology and behaviour (e.g., by increasing egg production and food intake) in response to male-derived peptides acquired during mating (*Carvalho et al., 2006*; *Barnes et al., 2008*). The synthesis of juvenile hormone (JH) in the *corpus allatum*, an endocrine gland, can be stimulated ex vivo by the male-derived Sex Peptide, suggesting regulation by mating (*Moshitzky et al., 1996*). Using rapid direct analysis in real time (DART) mass spectrometry, we profiled haemolymph of both virgin and mated female flies and established that the levels of in vivo circulating JH are indeed increased after mating (*Figure 3A*). The levels of two other juvenoid compounds, JH3-bisepoxide (JH3B) and methylfarnesoate (MF), were too low to be detected. We cannot, however, rule out that they also are regulated by mating and contribute to signalling through the JH pathway (*Yin et al., 1995*; *Tiu et al., 2012*; *Wen et al., 2015*). JH has been shown to sustain ovarian maturation through pleiotropic actions on adipose and reproductive tissues (*Flatt et al., 2005*), but its intestinal roles remain to be established. Consistent with a possible intestinal role, we detected transcript upregulation of the JH target *Kruppel homolog 1* (*Kr-h1*) (*Jindra et al., 2013*) in guts of mated females (*Figure 3—figure supplement 1*). To explore the roles of JH signalling on intestinal remodelling, we first fed methoprene, a JH analogue (JHa) (*Cerf and Georghiou, 1972*), to virgin female flies. This led to effects on intestinal progenitors, gut diameter, and lipid metabolism comparable to those triggered by mating (*Figure 3B–E,H*, *Figure 3—figure supplement 1*). We next blocked endogenous JH production by mis-expressing the protein phosphatase inhibitor *NiPp1* using the *corpus allatum*-specific driver *Aug21-Gal4* (*Siegmund and Korge, 2001*): a genetic manipulation known to result in adult-specific ablation of the *corpus allatum* and a dramatic reduction of JH titres in the haemolymph (*Yamamoto et al., 2013*). Depletion of systemic JH prevented mating-triggered remodelling: a phenotype that could be restored in these gland-ablated flies by JHa feeding (*Figure 3F,I*).

## JH signals directly to adult intestinal progenitors and enterocytes via Met and Gce receptors

To establish the cellular targets of JH and its mode of action, we interfered with JH signalling in a cell-autonomous manner in the intestine. We used the *esgReDDM* system, based on the widely used *esg-Gal4* driver (*Micchelli and Perrimon, 2006*), to target both classes of intestinal progenitor cells (ISCs and EBs). We first confirmed that expression of RNAi transgenes against either of the two previously identified JH receptors *Methoprene-tolerant* (*Met*) or *germ cell-expressed bHLH-PAS* (*gce*) (*Jindra et al., 2013*; *Jindra et al., 2015*) resulted in a significant reduction in their transcript levels (*Figure 3—figure supplement 1*). We then confined expression of these RNAi transgenes against *Met*, *gce*, or their target *Kr-h1* to adult intestinal progenitors using *esgReDDM*. Downregulation of any of these three genes fully prevented both the proliferative response to mating and midgut re-sizing, whereas overexpression of *Kr-h1* led to mating-like responses in virgin females (*Figure 3G,J*,

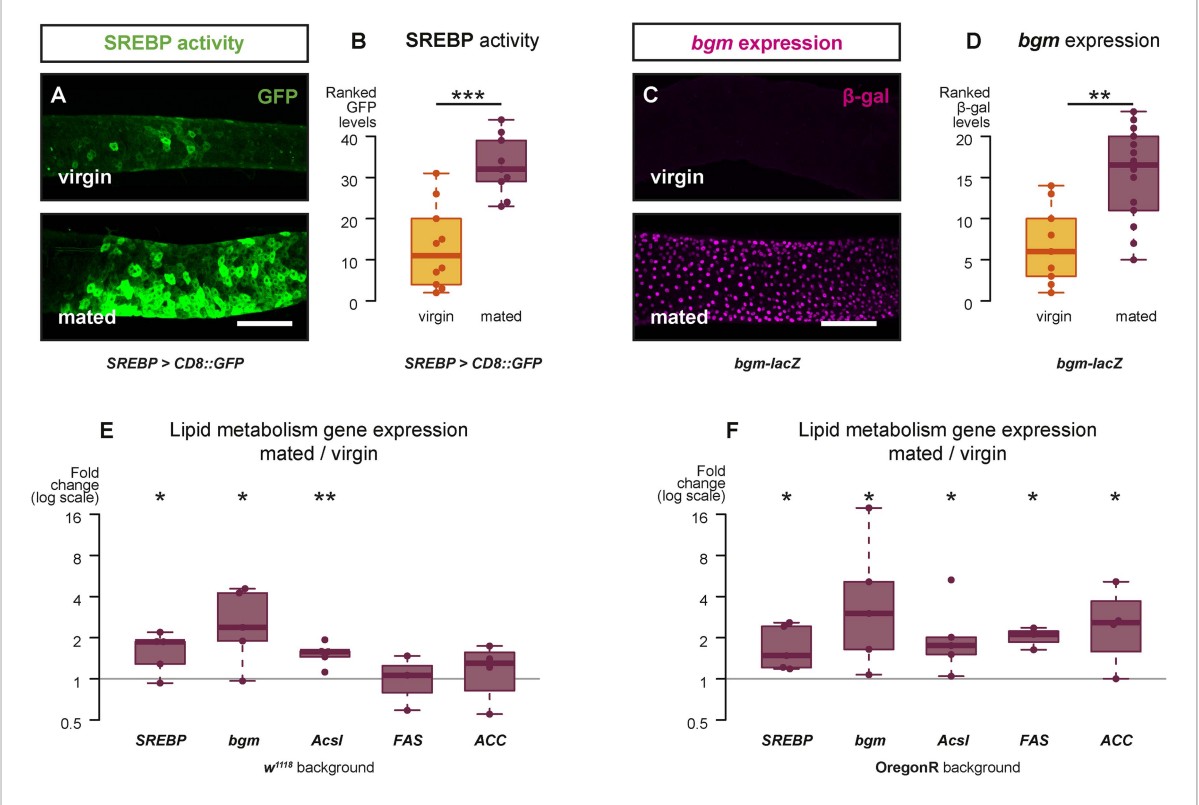

**Figure 2.** Mating changes the activity and/or expression of lipid metabolism genes in the intestine. At 3 days after mating, increased expression of a reporter that replicates the transcriptional regulation and post-translational modification of sterol regulatory element-binding protein (SREBP) is apparent in the posterior midgut (**A**, **A'**, quantified in **B**, p < 0.001, Mann–Whitney test). A *bgm* transcriptional reporter is also increased specifically in the ECs of the posterior midgut following mating (**C**, **C'**, quantified in **D**, p = 0.002, Mann–Whitney test). Transcript abundance of *SREBP*, *bgm*, and the *SREBP* targets *Acyl-CoA synthetase long-chain (Acsl)*, *Fatty acid synthase (FAS)*, and *Acetyl-CoA carboxylase (ACC)* is increased by mating in either one or both of the $w^{1118}$ and OregonR backgrounds (**E** $w^{1118}$: p = 0.02 *SREBP*, p = 0.02 *bgm*, p = 0.005 *Acsl*, p = 0.5 *FAS*, p = 0.3 *ACC*; **F** OregonR: p = 0.02 *SREBP*, p = 0.03 *bgm*, p = 0.03 *Acsl*, p = 0.01 *FAS*, p = 0.04 *ACC*, paired one-tailed t-test). See *Table 1* for full genotypes.

*Figure 3—figure supplement 1*), indicating direct actions of JH on intestinal progenitors. Intestinal progenitors with downregulated JH receptors were found in numbers comparable to those of controls in virgin females and were able to increase their proliferation in response to a JH-unrelated stimulus: the ROS-inducing compound paraquat (*Biteau et al., 2008*) (*Figure 3—figure supplement 1*). This indicates that they remain competent to respond to the well-studied homeostatic machinery that maintains gut integrity (*Jiang and Edgar, 2012*), and suggests that mating- and damage-induced proliferative mechanisms may differ and can be uncoupled. In ECs, targeted by the specific driver *Mex-Gal4* (*Phillips and Thomas, 2006*), only downregulation of *gce* strongly reduced the mating-dependent upregulation of a *bgm* reporter (*Figure 3K*, *Figure 3—figure supplement 1*). Together, these findings show that intestinal remodelling results from a rise in systemic JH triggered by mating. JH signals directly to intestinal progenitors to yield a larger organ in a *Met* and *gce*-dependent manner. Acting predominantly through *gce*, JH also adjusts gene expression in ECs to support increased lipid metabolism.

## Mating-triggered intestinal remodelling enhances reproductive output

Intestinal remodelling during reproduction could result from increased nutrient intake (*O'Brien et al., 2011*) or utilisation by the developing offspring. Alternatively, it may occur in preparation for, but be uncoupled from, such nutritional demands. Consistent with the latter idea, the mating-triggered changes in proliferation, midgut size, and SREBP activity are all still apparent in sterile female $ovo^{D1}$

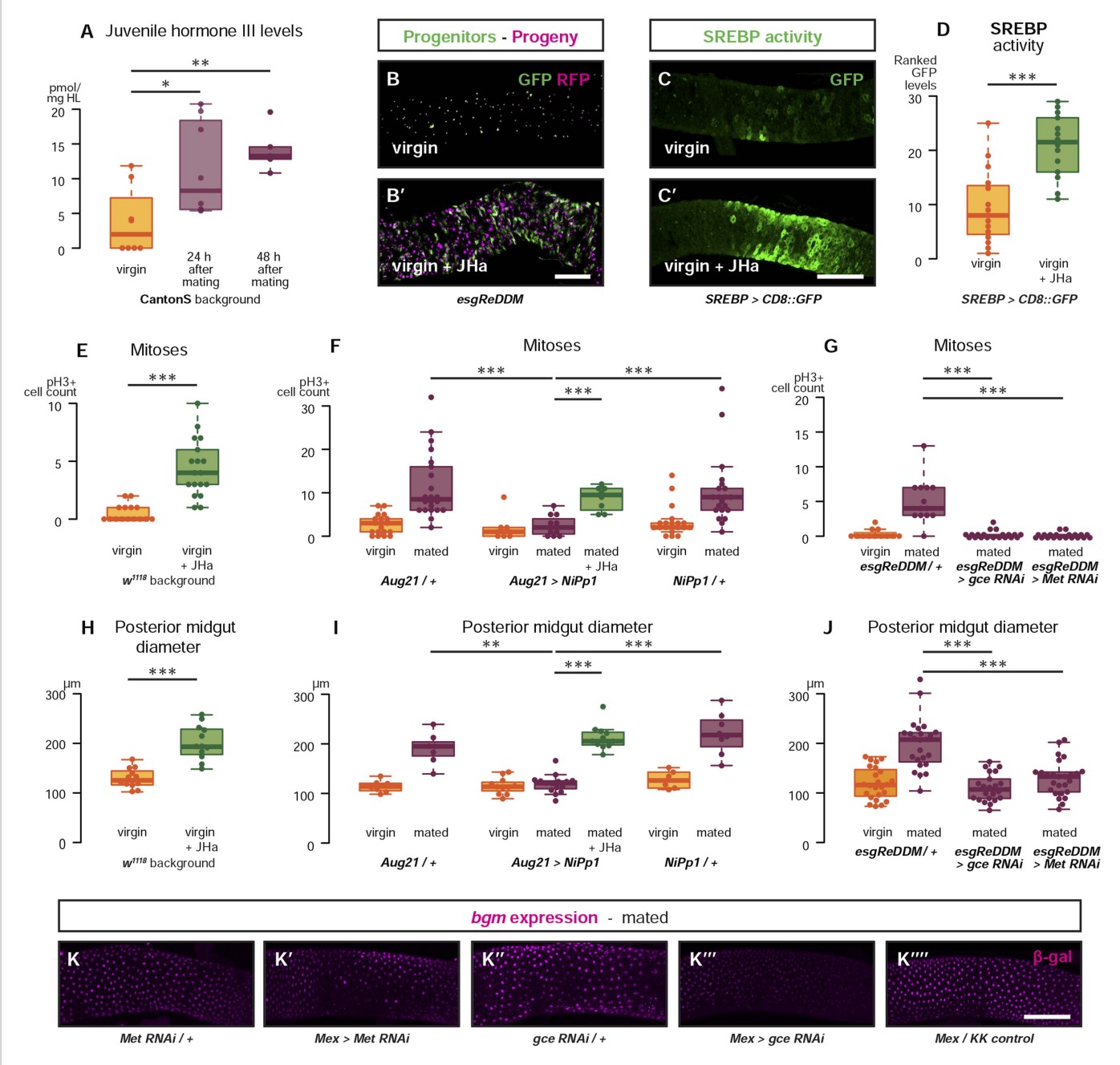

**Figure 3**. Systemic JH secreted after mating acts directly in the intestinal epithelium to drive reproductive remodelling. Circulating juvenile hormone (JH) is elevated after mating in the haemolymph of female flies (**A**, p = 0.02 at 24 hr, p = 0.002 at 48 hr, t-test with Holm's correction). Increased tissue renewal (**B**, **B'**) and SREBP activation (**C**, **C'**, quantified in **D**, p < 0.001, Mann–Whitney test) are apparent following a 3-day dietary supplementation with JH analogue (JHa). JHa treatment is sufficient to increase mitoses (**E**, p < 0.001, negative binomial GLM) and size (**H**, p < 0.001, t-test) of the posterior midgut. Conversely, when the endogenous JH source is genetically ablated by means of *Aug21 > NiPp1* (*Yamamoto et al., 2013*), the proliferation and size increase that follow mating are abolished, although they can be reinstated by feeding JHa (proliferation **F**, p < 0.001 between *Aug21/+* and *Aug21 > NiPp1* mated, p < 0.001 between *Aug21 > NiPp1* and *NiPp1/+* mated, p < 0.001 between *Aug21 > NiPp1* and *Aug21 > NiPp1* + JHa mated; all relevant comparisons between virgins are not significant, negative binomial GLM with Holm's correction; gut diameter **I**, p = 0.002 between *Aug21/+* and *Aug21 > NiPp1* mated, p < 0.001 between *Aug21 > NiPp1* and *NiPp1/+* mated, p < 0.001 between *Aug21 > NiPp1* and *Aug21 > NiPp1* + JHa mated; all relevant comparisons between virgins are not significant, t-test with Holm's correction). Downregulation of either *gce* or *Met* in adult progenitors abrogates post-mating proliferation (**G**, p < 0.001 between *esgReDDM/+* and *esgReDDM > gce RNAi* mated, p < 0.001 between *esgReDDM/+* and *esgReDDM > Met RNAi* mated, negative binomial GLM with Holm's correction) and gut size increase (**J**, p < 0.001 between *esgReDDM/+* and *esgReDDM > gce RNAi* mated, p < 0.001 between *esgReDDM/+* and *esgReDDM > Met RNAi* mated, t-test with Holm's correction). The upregulation of *bgm*

*Figure 3. continued on next page*

Figure 3. Continued

reporter upon mating is abolished by the downregulation of *gce*, but not *Met*, in ECs using the EC-specific driver *Mex-Gal4* (**K–K′′′′**). See *Table 1* for full genotypes.

The following figure supplement is available for figure 3:

**Figure supplement 1**. Intestinal JH signalling is relayed through Kr-h1 and underlies mating-dependent intestinal growth and gene expression phenotypes.

mutant flies in which egg production is blocked prior to vitellogenesis and which do not increase food intake after mating (*Barnes et al., 2008*) (*Figure 4—figure supplement 1*). To investigate the significance of intestinal remodelling, we used several RNAi transgenes to downregulate either the JH receptors or *SREBP*, which is activated by mating, specifically in adult ECs. In all cases, EC-specific downregulation led to a reduction in the number (but not viability) of eggs produced (*Figure 4E,F* and *Figure 4—figure supplement 1*), indicating that JH signalling is required to specifically enhance the quantity (fecundity), but not the quality (viability), of reproductive output. Progenitor cell-specific downregulation may also be expected to reduce fecundity; however, we detected expression of several intestinal progenitor drivers outside the intestine (data not shown), which could affect egg production independently of the intestine. More specific tools will be necessary to resolve this important issue.

The anatomical proximity between the ovary and the posterior midgut region where changes in lipid gene expression and activity take place (*Figure 4A,D*) raises the intriguing possibility that enhanced nutrient delivery from the intestine to the ovary may occur locally, to maximise loading into eggs. As the trafficked nutrients would be therefore released in the form of eggs, we used sterile $ovo^{D1}$ females to quantify lipid content, reasoning that it might accumulate in gut and/or peripheral fat stores in the absence of the local ovarian sink (*Parra-Peralbo and Culi, 2011*). Consistent with this idea, sterile $ovo^{D1}$ female flies accumulate peripheral fat after mating (*Figure 4B,C*), and lipid accumulation in the posterior midgut could be induced in fertile flies by either treatment of these sterile flies with JHa or by knocking down lipid shuttling proteins acutely, thereby blocking all lipid circulation (*Palm et al., 2012*) (*Figure 4G–J*). Together, these data show that the metabolic reprogramming of ECs by JH supports fecundity, thus confirming that intestinal plasticity is required to sustain reproductive output at the time of peak fertility. The importance of intestinal lipogenesis is becoming increasingly recognised in both flies and mammals (*Lodhi et al., 2011*; *Palm et al., 2012*; *Sieber and Thummel, 2012*; *Song et al., 2014*), and here we show that it underpins reproductive output. Notably, upregulation of *SREBP* target genes has been reported in the small intestine of lactating rats (*Athippozhy et al., 2011*), suggesting that our findings may be conserved beyond insects.

## Discussion

### Intestinal remodelling and the costs of reproduction

The onset of reproduction involves a significant shift in metabolic demands, now routed towards the growing offspring as well as the mother. *Drosophila* may experience a particularly extreme example of this shift after mating: an event that enhances egg production tenfold and triggers multiple metabolic and behavioural adaptations (*Kubli, 2003*; *Rogina et al., 2007*; *Avila et al., 2011*; *Cognigni et al., 2011*). These changes are in large part brought about by signals delivered by the male during copulation, in particular the Sex Peptide (*Kubli, 2003*; *Avila et al., 2011*). Several reports connect SP to the *corpus allatum* and JH production (*Moshitzky et al., 1996*; *Bontonou et al., 2015*), suggesting that the systemic effects of mating via SP could be carried out through this pathway. Intriguingly, both JH knockdown in females (*Yamamoto et al., 2013*) and SP deficiency in males (*Wigby and Chapman, 2005*) extend female lifespan while reducing reproductive output and/or peak fertility. This 'cost of mating' on lifespan is not relieved by sterility (*Ueyama and Fuyama, 2003*), suggesting that physiological effects in non-reproductive tissues are responsible. The intestinal reprogramming that we describe here represents a novel physiological target of postmating plasticity ideally placed at the interface between nutrition and reproduction. Ageing in flies is accompanied by reduced fertility

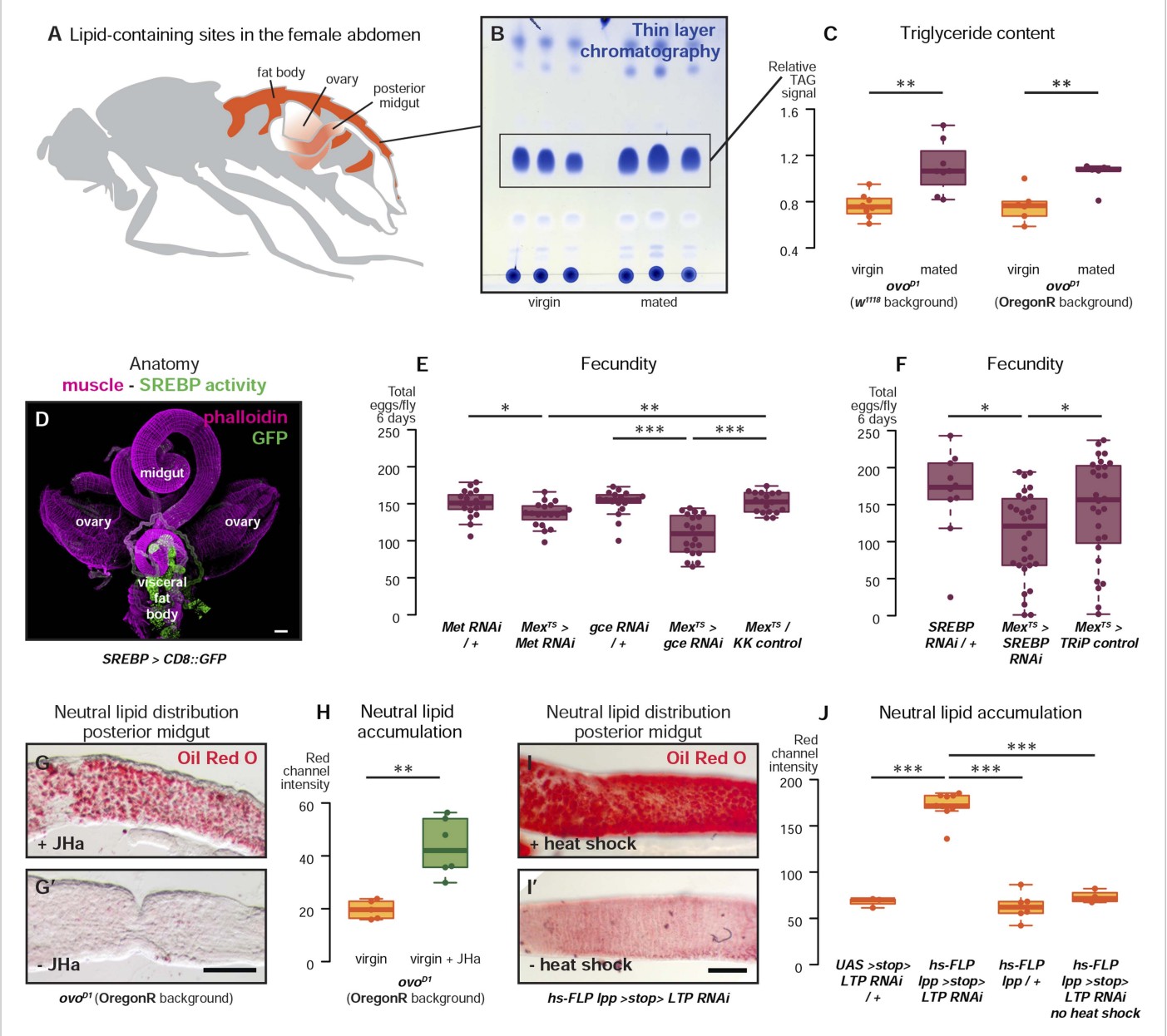

**Figure 4**. Metabolic remodelling of ECs by JH sustains reproduction. Lipid-harbouring tissues (fat body, posterior midgut, and ovary) are found in close proximity in the fly's abdomen (represented schematically in **A**, and in confocal microscopy in **D**). The amount of stored triglycerides (TAG) in the carcass of 3-day mated sterile female flies is increased compared to virgins (**B**, p = 0.003 in $w^{1118}$, p = 0.009 in OregonR, t-test), as quantified by thin-layer chromatography (**C**). Adult-specific downregulation of JH receptors gce and Met or SREBP in ECs reduces the total progeny produced by females in the 6 days following their first mating (**E**, p = 0.01 between Met RNAi/+ and $Mex^{ts}$ > Met RNAi, p = 0.007 between $Mex^{ts}$ > Met RNAi and $Mex^{ts}$/KK control, p < 0.001 between gce RNAi/+ and $Mex^{ts}$ > gce RNAi, p = 0.007 between $Mex^{ts}$> gce RNAi and $Mex^{ts}$/KK control; **F** p = 0.04 between SREBP RNAi/+ and $Mex^{ts}$ > SREBP RNAi, p = 0.04 between $Mex^{ts}$ > SREBP RNAi and $Mex^{ts}$/TRiP control, t-test with Holm's correction). In the absence of the ovarian lipid sink in sterile $ovo^{D1}$ virgin flies, treatment with JHa increases neutral lipid content, as revealed by Oil Red O staining, in the posterior midgut (**G**, **G′**, quantified in **H**: p = 0.002, t-test). Acute block of lipid export by heat-shock activation of lpp > stop > LTP RNAi (**Palm et al., 2012**) in virgin females results in heavy accumulation of neutral lipid in this gut region, further indicating that this midgut region provides a net source of lipid in adult flies (**I**, quantified in **J**: p < 0.001 between LTP RNAi/+ and lpp > stop > LTP RNAi, p < 0.001 between lpp > stop > LTP RNAi and lpp > stop>/+, p < 0.001 between lpp > stop > LTP RNAi and lpp > stop > LTP RNAi heat shock control, t-test with Holm's correction). See **Table 1** for full genotypes.

The following figure supplement is available for figure 4:

**Figure supplement 1**. Reproductive intestinal remodelling is uncoupled from germline demands and is needed to sustain reproduction.

(*Economos et al., 1979*) and intestinal dysplasia (*Biteau et al., 2008*; *Choi et al., 2008*), and genetic manipulations that affect intestinal progenitors can affect lifespan (*Biteau et al., 2010*; *Rera et al., 2011*). Thus, it will be informative to explore the links between JH-triggered postmating responses in lifespan, age-dependent intestinal and reproductive dysfunction, and lifetime fertility.

## Hormonal remodelling of adult organs

In larvae, the *corpora allata* integrate age and metabolic status information to optimise developmental progression (*Mirth and Shingleton, 2012*; *Sarraf-Zadeh et al., 2013*). Increasing evidence is revealing that, in adults, this insect endocrine organ acts as a nexus that detects changes in the organism's circumstances and condition to alter its metabolic and/or reproductive set points. It does so through regulated release of JH, with pleiotropic effects on ovarian maturation, adipose tissue, learning and memory, diapause, innate immunity, and ageing (*Nijhout and Riddiford, 1974*; *Jowett and Postlethwait, 1980*; *Fahrbach and Robinson, 1996*; *Wyatt and Davey, 1996*; *Flatt et al., 2005*; *Riddiford, 2012*; *Yamamoto et al., 2013*). Some of these effects may be modulated by crosstalk between JH and other systemic signals such as insulin-like peptides and ecdysteroids (*Jindra et al., 2013*; *Koyama et al., 2013*; *Mirth et al., 2014*; *Rauschenbach et al., 2014*), but the cellular and molecular targets of JH action remain incompletely understood. Our findings have uncovered a direct and functionally significant effect on adult organ plasticity by showing that JH promotes proliferation and resets gut size through its actions on intestinal progenitors and activates expression of lipid metabolism genes in ECs. Based on structural and functional similarities, insect JH has been compared to mammalian thyroid hormones (*Flatt et al., 2006*): key energy balance regulators often associated with gastrointestinal alterations when pathologically dysregulated (*Middleton, 1971*). Given the well-established changes in thyroid function during human pregnancy (*Glinoer, 1997*), it will therefore be of interest to explore their contribution to reproductive intestinal remodelling. Downstream of its receptor(s), relay of JH signalling in the intestine may differ from the classical model in which *Met* and *gce* act redundantly (*Abdou et al., 2011*; *Jindra et al., 2013*). Indeed, downregulation of either gene alone is sufficient to prevent the mating-induced changes in intestinal progenitors: a finding that we confirmed by observing that viable *Met*[27] mutants also fail to undergo mating-induced remodelling (data not shown). The actions of Met and Gce may also be cell-specific, as suggested by a preferential requirement for *gce* in ECs, and may result from different Met/gce expression levels (our unpublished observations) and/or interacting partners. Candidates to consider include Taiman, homologous to the mammalian steroid receptor coactivator 1 (SRC-1)/NCoA-1/p160 (*Charles et al., 2011*; *Li et al., 2011*; *Zhang et al., 2011*) and, more intriguingly, circadian clock proteins: Met-binding partners recently shown to coordinate the switch from diapause to reproduction in other insects (*Shin et al., 2012*; *Bajgar et al., 2013*).

## Adult organ plasticity, obesity, and cancer

Adult organ plasticity is not a peculiarity of *Drosophila* reproduction; examples of changes in intestinal size and nutrient utilisation are widespread across the animal kingdom in response to both environmental and internal challenges (*Carey, 1990*; *Hammond, 1997*; *Piersma and Lindstrom, 1997*; *Speakman, 2008*; *O'Brien et al., 2011*). Although intestinal remodelling has not been assessed in human pregnancy, it could be one of the major drivers for the changes in gut microbiota observed during pregnancy (*Koren et al., 2012*) and could contribute to changes in gastrointestinal physiology, common during pregnancy (*Keller et al., 2008*). Resetting of anatomical or metabolic features of internal organs may thus be a broadly used strategy to achieve a positive energy balance which, when matched to the developing offspring's demands, will contribute to reproductive success. However, if deployed in the absence of such demands, organ remodelling could contribute to the weight gain and increased fat mass that has been observed upon gonadectomy of multiple species including mice, rats, cats, monkeys, and other mammals (*Hansen et al., 2013* and references therein). In a more physiological context, inappropriate persistence of such metabolic remodelling beyond pregnancy and lactation could similarly contribute to post-pregnancy weight retention in humans—a phenotype that, at least in mice, is correlated with enhanced intestinal function (*Casirola and Ferraris, 2003*; *Gore et al., 2003*). Similarly, inappropriate persistence of JH-like mechanisms that change the homeostatic set point of adult stem cells and their progeny to transform an organ may also help explain why pregnancy changes the susceptibility to certain cancers (*Gwinn et al., 1990*).

# Materials and methods

## Fly strains

For wild-type experiments, the genetic backgrounds *w^1118^*, OregonR, and CantonS were used as indicated in the figures and/or full genotypes list (*Table 1*). The following transgenic and mutant stocks were used: *esg-Gal4* (Bloomington, unknown insertion), *tub-Gal80* (Bloomington 7018, *McGuire et al., 2003*), *UAS-mCD8::GFP* (Bloomington 5130, *Lee and Luo, 1999*), *UAS-H2B::RFP* (presumed from *Langevin et al., 2005*), *caudal-Gal4* (insertion used in *Ryu et al., 2008*), *SREBP-Gal4* (Bloomington 38395, *Kunte et al., 2006*), *bgm-lacZ* (Bloomington 28120, *Min and Benzer, 1999*), *Aug21-Gal4* (Bloomington 30137, *Siegmund and Korge, 2001*), *UAS-NiPp1* (Bloomington 23712, *Parker et al., 2002*), *tub-Gal4* (Bloomington 5138, *Lee and Luo, 1999*), *Mex-Gal4* (*Phillips and Thomas, 2006*), *UAS-Kr-h1* (DGRC 120052, referred to as *UAS-Kr-h1*), *ovo^D1^* (*Busson et al., 1983*), *hs-FLP; lpp-Gal4* and *UAS > stop > LTP RNAi* stocks (both from *Palm et al., 2012*). RNAi constructs were obtained from VDRC for *gce* (KK101814, GD11178 and GD47465), *Met* (KK100638), *Kr-h1* (KK107935), and *SREBP* (GD37641 and GD37640), as well as the genetically matched KK control (KK60100); and from the Bloomington TRiP collection for *SREBP* (34073) and the genetically matched TRiP control (GFP in valium10, 35786). Because the control stocks are generated in the same background as the *RNAi* lines used, the *Gal4* parental control (e.g., *yv; Mex-Gal4/+; tub-Ga80^ts^/UAS-GFP*) is genetically matched to the experimental genotype (e.g., *yv; Mex-Gal4/+; tub-Gal80^ts^/UAS-SREBP RNAi TRiP*). The line referred to as *UAS-Kr-h1_GS* is *GS(2)73ES2b*, which was isolated in a genetic screen for enhancer/suppressors of a large-eye phenotype caused by Dl overexpression in the Dominguez lab. Genomic DNA flanking the P-element insertions in the *GS(2)73ES2b* stock were recovered by inverse PCR and sequenced. A BLAST search with the obtained sequence produced perfect matches to the genomic region upstream of the *Kr-h1* gene (26B5 Chromosome 2L: 6,082,603,...,6,096,498).

## Fly husbandry

Fly stocks were reared on a standard cornmeal/agar diet (5.5% cornmeal, 6% dextrose, 1.3% yeast, 0.55% agar supplemented with 0.18% nipagin and 2.9 ml/l propionic acid) or 'Iberian' diet (4.4% wheat flour, 6% brown sugar, 3% yeast, 1% agar supplemented with 0.04% nipagin and 7.6 ml/l of propionic acid). All experimental flies were kept at 25°C expect for those containing temperature-sensitive regulation (*tub-Gal80^ts^*), which were set up at 18°C (restrictive temperature) and transferred to 29°C (permissive temperature) at the time when activation was needed in the specific experiment. For all experiments, experimental and control flies were handled in parallel and experienced the same temperature shifts and treatments.

For the analysis of mating and JHa effects, virgin female flies were collected at eclosion, aged for 4–5 days on standard food and then transferred for 3 days (7 days for flies harbouring the *esgReDDM* transgenes, as these flies show a delay in mating responses at 29°C) into new tubes in the presence of wild-type males (typically 4–5 females + 6 males) or food supplemented with 1.5 mM methoprene (Sigma-Aldrich, St Louis, MO, PESTANAL 33375, racemic mixture), added to freshly prepared food when still liquid but <50°C. This concentration was chosen in a pilot dilution test from 0.5 to 7.5 mM as the one that induced activation of the *SREBP-Gal4* reporter to levels comparable to mating, and corresponds to approximately half of the concentration used in a previous study (*Flatt and Kawecki, 2007*). Controls were age-matched virgin females, also transferred to new tubes for the same time but without the addition of males and/or methoprene.

For the paraquat experiments, virgin female flies were raised at 18°C and aged for 4–5 days after eclosion, at which point they were starved for 4 hr without water. The flies were then transferred to vials containing filter paper soaked in 5% sucrose with or without 10 mM paraquat dichloride (Sigma-Aldrich). After spending 24 hr at 29°C in these vials, their midguts were dissected and stained for pH3 as described before.

## Antibodies

The following antibodies were used: rabbit anti-pH3 (1:2000, Upstate, Merck-Millipore, Germany), sheep anti-GFP (1:1000, Biogenesis, for *esgReDDM* staining), goat anti-GFP (1:1500, Abcam, UK, for *SREBP > mCD8::GFP* staining), rabbit anti-β-galactosidase (1:5000, MP Biomedicals, Santa Ana, CA); secondary antibodies were either FITC/Cy3 conjugates from Jackson ImmunoResearch (1:200, West

**Table 1.** Full genotypes

| Genotype in text/figure | Full genotype | Figure panel(s) |
|---|---|---|
| esgReDDM | w; esg-Gal4, UAS-mCD8::GFP/+; tub-Gal80$^{ts}$, UAS-H2B::RFP/+; + | *Figure 1A, Figure 1—figure supplement 1C,D, Figure 3B* |
| w$^{1118}$ background | w$^{1118}$; +; +; + | *Figure 1B–D, Figure 2E, Figure 3E,H, Figure 3—figure supplement 1H, Figure 4—figure supplement 1A,B* |
| OregonR background | +; +; +; + | *Figure 1E, Figure 1—figure supplement 1A,B, Figure 2F, Figure 3—figure supplement 1H* |
| caudal > H2B::RFP | w; caudal-Gal4/+; UAS-H2B::RFP/+; + | *Figure 1F, Figure 3—figure supplement 1A* |
| SREBP > CD8::GFP | w/+; SREBP-Gal4/+; UAS-CD8::GFP/+; + | *Figure 2A,B, Figure 3C,D, Figure 4D, Figure 4—figure supplement 1D* |
| bgm-lacZ | w/+; bgm-lacZ/+; +; + | *Figure 2C,D* |
| CantonS background | +; +; +; + | *Figure 3A* |
| Aug21/+ | w; Aug21-Gal4/+; +; + | *Figure 3F,I* |
| Aug21 > NiPp1 | w; Aug21-Gal4/+; UAS-NiPp1/+; + | *Figure 3F,I* |
| NiPp1/+ | w; +; UAS-NiPp1/+; + | *Figure 3F,I* |
| esgReDDM/+ | w; esg-Gal4, UAS-mCD8::GFP/+; tub-Gal80$^{ts}$, UAS-H2B::RFP/+; + | *Figure 3G,J, Figure 3—figure supplement 1D,F,G,I,J* |
| esgReDDM > gce RNAi | w; esg-Gal4, UAS-mCD8::GFP/UAS-gce RNAi KK101814; tub-Gal80$^{ts}$, UAS-H2B::RFP/+; + | *Figure 3G,J, Figure 3—figure supplement 1C–G* |
| esgReDDM > Met RNAi | w; esg-Gal4, UAS-mCD8::GFP/UAS-Met RNAi KK100638; tub-Gal80$^{ts}$, UAS-H2B::RFP/+; + | *Figure 3G,J* |
| esgReDDM > Kr-h1 RNAi | w; esg-Gal4, UAS-mCD8::GFP/UAS-Kr-h1 RNAi KK107935; tub-Gal80$^{ts}$, UAS-H2B::RFP/+; + | *Figure 3—figure supplement 1I,J* |
| esgReDDM > Kr-h1$_{GS}$ | w; esg-Gal4, UAS-mCD8::GFP/UAS-Kr-h1$_{GS}$;tub-Gal80$^{ts}$, UAS-H2B::RFP/+; + | *Figure 3—figure supplement 1I* |
| esgReDDM > Kr-h1$_{UAS}$ | w; esg-Gal4, UAS-mCD8::GFP/UAS-Kr-h1$_{UAS}$; tub-Gal80$^{ts}$, UAS-H2B::RFP/+; + | *Figure 3—figure supplement 1I,J* |
| tub$^{ts}$> Met RNAi | w; tub-Gal80$^{ts}$/UAS-Met RNAi KK100638; tub-Gal4, UAS-mCD8::GFP/+; + | *Figure 3—figure supplement 1B* |
| tub$^{ts}$ > gce RNAi | w; tub-Gal80$^{ts}$/UAS-gce RNAi KK101814; tub-Gal4, UAS-mCD8::GFP/+; + | *Figure 3—figure supplement 1B* |
| tub$^{ts}$ > Kr-h1 RNAi | w; tub-Gal80$^{ts}$/UAS-Kr-h1 RNAi KK107935; tub-Gal4, UAS-mCD8::GFP/+; + | *Figure 3—figure supplement 1K* |
| tub$^{ts}$ > SREBP RNAi GD | w; tub-Gal80$^{ts}$/UAS-SREBP RNAi GD37640; tub-Gal4, UAS-mCD8::GFP/+; + | *Figure 3—figure supplement 1K* |
| tub$^{ts}$ > SREBP RNAi TRiP | w; tub-Gal80$^{ts}$ +; tub-Gal4, UAS-mCD8::GFP/UAS-SREBP RNAi TRiP34073; + | *Figure 3—figure supplement 1K* |
| tub$^{ts}$/+ | w; tub-Gal80$^{ts}$/+; tub-Gal4, UAS-mCD8::GFP/+; + | *Figure 3—figure supplement 1B,K* (control) |
| Met RNAi/+ | w; bgm-lacZ/UAS-Met RNAi KK100638; +; + | *Figure 3K, Figure 3—figure supplement 1L* |
| Mex > Met RNAi | w; Mex-Gal4, bgm-lacZ/UAS-Met RNAi KK100638; +; + | *Figure 3K, Figure 3—figure supplement 1L* |
| gce RNAi/+ | w; bgm-lacZ/UAS-gce RNAi KK101814; +; + | *Figure 3K, Figure 3—figure supplement 1L* |
| Mex > gce RNAi | w; Mex-Gal4, bgm-lacZ/UAS-gce RNAi KK101814; +; + | *Figure 3K, Figure 3—figure supplement 1L* |
| Mex/KK control | w; Mex-Gal4, bgm-lacZ/attp40; +; + | *Figure 3K, Figure 3—figure supplement 1L* |
| ovo$^{D1}$ (w$^{1118}$ background) | w$^{1118}$; +; ovo$^{D1}$/+; + | *Figure 4C, Figure 4—figure supplement 1A,B* |

*Table 1. Continued on next page*

*Table 1. Continued*

| Genotype in text/figure | Full genotype | Figure panel(s) |
|---|---|---|
| ovo^D1 (OregonR background) | +/w^1118; +; ovo^D1/+; + | *Figure 4C,G,H* |
| Met RNAi/+ | w; UAS-Met RNAi KK100638/+; +; + | *Figure 4E, Figure 4—figure supplement 1H* |
| Mex^ts> Met RNAi | w; Mex-Gal4/UAS-Met RNAi KK100638; tub-Gal80^ts/+; + | *Figure 4E, Figure 4—figure supplement 1H* |
| gce RNAi/+ | w; UAS-gce RNAi KK101814/+; +; + | *Figure 4E, Figure 4—figure supplement 1H* |
| Mex^ts > gce RNAi | w; Mex-Gal4/UAS-gce RNAi KK101814; tub-Gal80^ts/+; + | *Figure 4E, Figure 4—figure supplement 1H* |
| Mex^ts/KK control | w; Mex-Gal4/attp40; tub-Gal80^ts/+; + | *Figure 4E, Figure 4—figure supplement 1H* |
| SREBP RNAi/+ | w/y, v; +; UAS-SREBP RNAi 34073; + | *Figure 4F* |
| Mex^ts> SREBP RNAi | w/y, v; Mex-Gal4/+; tub-Gal80^TS/UAS-SREBP RNAi 34073; + | *Figure 4F* |
| Mex^ts/TRiP control | w/y, v; Mex-Gal4/+; tub-Gal80^ts/UAS-GFP; + | *Figure 4F* |
| hs-FLP lpp > stop > LTP RNAi | w, hs-FLP/w; lpp-Gal4/+; UAS > stop > LTP RNAi/+; + | *Figure 4I,J* |
| UAS > stop > LTP RNAi/+ | w; +; UAS > stop > LTP RNAi/+; + | *Figure 4J* |
| hs-FLP lpp/+ | w, hs-FLP/w; lpp-Gal4/+; +; + | *Figure 4J* |
| esgReDDM/ovo^D1 | ovo^D1/w; esg-Gal4, UAS-mCD8::GFP/+; tub-Gal80^ts, UAS-H2B::RFP/+; + | *Figure 4—figure supplement 1C* |
| SREBP > CD8::GFP/ovo^D1 | w/+; SREBP-Gal4/+; UAS-CD8::GFP/ovo^D1; + | *Figure 4—figure supplement 1D,E* |
| gce RNAi GD11178/+ | w; UAS-gce RNAi GD11178; +; + | *Figure 4—figure supplement 1F* |
| Mex^ts> gce RNAi GD11178 | w; Mex-Gal4/UAS-gce RNAi GD11178; tub-Gal80^ts/+; + | *Figure 4—figure supplement 1F* |
| gce RNAi GD47465/+ | w; +; UAS-gce RNAi GD47465/+; + | *Figure 4—figure supplement 1F* |
| Mex^ts> gce RNAi GD47465 | w; Mex-Gal4/+; tub-Gal80^ts/UAS-gce RNAi GD47465; + | *Figure 4—figure supplement 1F* |
| Mex^ts/+ | w; Mex-Gal4/+; tub-Gal80^ts+; + | *Figure 4—figure supplement 1F* |
| SREBP RNAi GD37641/+ | w; UAS-SREBP RNAi GD37641/+; +; + | *Figure 4—figure supplement 1G,I* |
| Mex^ts> GD37641 | w; Mex-Gal4/UAS-SREBP RNAi GD37641; tub-Gal80^ts/+; + | *Figure 4—figure supplement 1G,I* |
| SREBP RNAi GD37640/+ | w; UAS-SREBP RNAi GD37640/+; +; + | *Figure 4—figure supplement 1G,I* |
| Mex^ts> RNAi GD37640 | w; Mex-Gal4/+; tub-Gal80^ts/UAS-SREBP RNAi GD37640; + | *Figure 4—figure supplement 1G,I* |

Grove, PA, for *SREBP > mCD8::GFP* and *bgm-lacZ*) or Alexa488/647 conjugates from Invitrogen Life Technologies (1:1000, Carlsbad, CA, for *esgReDDM* and *caudal > H2B::RFP*). Preparations for proliferation analysis were counterstained with DAPI (Sigma-Aldrich) and mounted in Fluoromount-G (Southern Biotech, Birmingham, AL). Preparations for reporter analysis were mounted in Vectashield with DAPI (Vector Labs, Burlingame, CA).

## Proliferation and size quantifications

Quantification of mitoses in wild-type and *ovo^D1* female flies was carried out by counting individual nuclei marked by the mitotic marker pH3 using a Nikon Eclipse 90i Fluorescence microscope through a 40× objective. For the acquisition of gut images in these samples, a single 1392 × 1040 field was acquired posterior to the midgut–hindgut boundary using QCapture software (QImaging). Progeny dynamics were analysed using the *esgReDDM* system (*Antonello et al., 2015*), which has the genetic makeup *esg-Gal4, UAS-mCD8::GFP; tub-Gal80^ts, UAS-H2B::RFP*. At the permissive temperature of 29°C, the GFP reporter is expressed in *esg-Gal4* positive cells (ISCs and EBs), but due to the perdurance of the RFP-tagged histone H2B::RFP the *esg-Gal4*-negative progeny (including ECs and EECs) generated from these progenitors since the shift to permissive temperature is additionally labelled in red. To restrict progeny analysis to mating-induced changes, *esgReDDM* flies were maintained at 18°C, such that Gal4 expression is suppressed by *tub-Gal80^ts*, and moved to 29°C only at the time of mating. After 3 days of mating at 29°C, guts were dissected and stained for GFP and pH3 (the endogenous RFP signal was

detected directly). EC number in the posterior midgut was assessed by imaging the entire gut of *caudal > H2B::RFP* flies and counting the number of RFP-marked cells. Confocal images were obtained with a Leica TCS SP5 inverted confocal microscope using a 20× air objective for *esgReDDM* and a 10× air objective for *caudal > H2B::RFP*. Stacks were typically collected every 1 µm, and the images (1024 × 1024) were reconstructed using maximum projection. Bright-field images or confocal maximum projections were loaded into ImageJ (*Schneider et al., 2012*) and the line tool used to quantify the width of the gut across the centre of the image. ImageJ was also used to outline the guts of *esgReDDM* flies using the polygon tool before analysing the resulting region of interest (ROI) with a custom MATLAB (The MathWorks, Inc.) script optimised for the ReDDM method. Extended details about this analysis are available from (*Antonello et al., 2015*). Briefly, maximum projections were adjusted for levels and offsets and filtered to remove noise (using always the same parameters for scans within one experiment), then the area of the gut was identified by background staining and the cell nuclei by DAPI signal. The size of nuclei can be used to discriminate between diploid cells (ISCs, EBs, and EECs) and polyploid ECs. The red-labelled nuclei (persistent H2B::RFP) and green-labelled cells (mCD8::GFP) were identified by segmentation and compared to the pattern of nuclei defined by DAPI to generate a report of total ECs (large DAPI cells), total progenitors + progeny (RFP signal), total ISCs and EBs (GFP signal), and total area. The same script was also used to count the number of *caudal > H2B::RFP* cells.

## Analysis of reporter gene expression

For *SREBP > mCD8::GFP* and *bgm-lacZ* experiments, confocal images were obtained with a Leica SP5 upright confocal microscope using a 20× glycerol immersion objective. A single 20× field (1024 pixels wide) immediately posterior to the midgut–hindgut boundary was acquired with a Z resolution of 1.5 µm. ImageJ was used to generate a maximum projection for each sample and all images pertaining to one experiment were loaded as separate layers into a single Adobe Photoshop CS6 file. The layers were then ranked blindly on the basis of their relative intensity in the relevant channel.

## qPCR

To quantify mating-induced changes in gene expression, posterior midguts from at least 10 adult female flies were dissected, discarding Malphigian tubules and the hindgut. To determine the knockdown efficiency of the RNAi transgenes, *tub-Gal80^ts^; tub-Gal4, UAS-GFP* was used to downregulate them ubiquitously. 8–10 third instar larvae were collected from crosses kept at 21°C and were shifted to 29°C for 3 hr to allow RNAi transgene expression. Samples (posterior midguts or whole larvae) were directly stored on dry ice and at −80°C in RNAlater TissueProtect Tubes (Qiagen, the Netherlands) until total RNA was extracted using RNeasy Mini Kit (Qiagen), from which cDNAs were prepared with SuperScript First-Strand Synthesis System (Invitrogen Life Technologies) using oligo-dT primers. Quantitative PCR was performed using the SYBR Green PCR Master Mix (Applied Biosystems Life Technologies) in a 7500 Real-Time PCR System (Applied Biosystems) using the housekeeping gene *rp49* as a control. All qPCRs were performed in triplicate and the relative expression was calculated using comparative Ct method.

Primers used:

|  | Forward 5′–3′ | Reverse 3′–5′ |
| --- | --- | --- |
| *SREBP* | GCAAAGTGCGTTGACATTAACC | AGTGTCGTGTCCATTGCGAA |
| *bgm* | GCAATCGATTTGCGTGACCA | GGCCCAGGACGATTGTAGAG |
| *Acsl* | CGGAGATCCGACAAAGCAGT | TGAGCACAGCTCCTCAAAGG |
| *FAS* | GACATTCGATCGACGCCTCT | GCTTTGGCTTCTGCACTGAC |
| *ACC* | AATTCTCCAAGGCTCGTCCC | CATGCCGCAATTGTTTTCGC |
| *Kr-h1* | ACAATTTTATGATTCAGCCACAACC | GTTAGTGGAGGCGGAACCTG |
| *gce* | AGCTGCGTATCCTGGACACT | TCGAGAGCTGAAACATCTCCAT |
| *Met* | CCGCCGTCCTTAGATTCGC | GTTCCCTTGAGGCCGGTTT |
| *rp49* | TGTCCTTCCAGCTTCAAGATGACCATC | CTTGGGCTTGCGCCATTTGTG |

## Detection of circulating JH hormone by DART-MS

Haemolymph was extracted from virgin or mated females using pulled glass microcapillary needles (10 µl vol, #2-000-010; Drummond Scientific, PA, USA). The needle tip was placed into the gap between the anepisternum and anepimeron of anesthetised flies, and haemolymph was collected using a slight vacuum (0.2–1.0 mPa) for ~30 s. Haemolymph from 45 to 50 flies was collected in the same needle. The contents were ejected into a 0.1 ml glass vial insert (Thermo Fisher Scientific, MA, USA) by applying pressurised air (~5–6 kPa) with a Femtojet microinjector (Eppendorf, NY, USA), and weighed prior to extraction. 20 µl of MeOH was added to the haemolymph followed by extraction with 20 µl of hexane, repeated four times. Pooled hexane extract was evaporated under a gentle stream of $N_2$ and reconstituted in 10 µl of hexane. All extracts were prepared and measured immediately after collection.

Mass spectra were acquired with an atmospheric pressure ionisation time-of-flight mass spectrometer (AccuTOF-DART, JEOL USA, Inc.) equipped with a DART interface and operated with a resolving power of 6000 (FWHM definition). The RF ion guide voltage was set at 600 V. The atmospheric pressure ionisation interface potentials were as follows: orifice 1 = 15 V, orifice 2 = 5 V, ring lens = 5 V. Mass spectra were stored at a rate of one spectrum per second with an acquired m/z range of 60–1000. The DART interface was operated in positive-ion mode using helium gas with the gas heater set to 200°C. The glow discharge needle potential was set to 3.5 kV. Electrode 1 was set to +150 V, and electrode 2 was set to +250 V. Helium gas flow was set to 2.0 l/min. Calibration for exact mass measurements was accomplished by acquiring a mass spectrum of polyethylene glycol (average molecular weight 600) as a reference standard in every data file. Analysis was done with JEOL MassCenter software (version 1.3.0.1). Accurate mass measures and isotope pattern matching by MassMountaineer (FarHawk Marketing Services, Ontario, CA) were used to support elemental composition assignments.

2 µl of the haemolymph hexane extract was placed on the tip of a borosilicate glass capillary. The capillary was introduced to the DART ion source with a micromanipulator, thus allowing for reproducible placement of the sample. Each extract was measured 4–5 times. The averaged signal intensity was normalised to the total weight of the haemolymph and converted to absolute quantities after establishing a calibration curve with a JHIII standard (Santa Cruz Biotechnology, CA, USA, CAS 24198-95-6). Analysis of JHIII by DART produces two signature ions at m/z 267.20 (intact molecule) and at m/z 249.18 (loss of water), consistent with a previous study (Navare et al., 2010). The abundance of the $[M-H_2O + H]^+$ signal peak was used for all measurements because the parent ion at m/z 267.20 could not be consistently resolved due to interference from other signals. To detect other juvenoid compounds, the following mass signatures were used: methylfarnesoate ($[M + H]^+$ 251.20) and JHIII Bisepoxide ($[M + H]^+$ 283.19). DART MS was previously shown to be an effective method for quantitative and high-sensitivity measurements of JHIII (Navare et al., 2010).

## Fecundity and egg viability experiments

Flies for fecundity and egg viability experiments were raised at 18°C to prevent the expression of the RNAi transgenes during development, then shifted to 29°C in late pupariation (after ~20 days). Virgin females were collected upon eclosion and after 4 days mated overnight to OregonR males (10 males, 10 females per vial). Males were then removed, individual female flies were transferred to a new single vial of yeast-supplemented standard food (cornmeal/agar diet with 5% yeast content) every 48 hr, and eggs were counted from the vacated vial to quantify fecundity. To assess egg viability, a fraction of the egg-containing vials were then maintained at 29°C, and the number of eclosed progeny was counted and compared with egg counts. Each genotype cross was performed three times, and egg production from each fly was assessed over three 48 hr repeats, covering a total of 6 days of egg laying.

## Thin-layer chromatography (TLC)

Ovaries and guts were removed from flies immobilised on ice and the remaining carcasses (three flies per sample) were immediately homogenised in a mixture of methanol (60 µl), chloroform (150 µl), and water (75 µl), following previously described procedures (Al-Anzi et al., 2009; Hildebrandt et al., 2011). After an extraction period (1 hr at 37°C), aqueous and organic phases were separated by the addition of a 1:1 mixture of 1 M potassium chloride and chloroform (75 µl each). Samples were briefly

centrifuged and 120 µl of the organic phase was transferred to fresh Eppendorf tubes and left to air-dry for 3 hr. The resulting dessicated lipids were resuspended in 16 µl of a 1:1 chloroform:methanol mixture. 3 µl of each sample was applied to TLC plates (Merck Millipore116487) and lipid species were separated by standing the plates in ~1 cm of a mobile phase consisting of 69.5% hexane, 29.5% diethyl ether, and 1% acetic acid. Once the mobile phase had traversed the plates, they were briefly dried and then dipped in a cerium-ammonium-molybdate stain (ammonium heptamolybdate tetrahydrate 2.5 g, cerium (IV) sulphate hydrate complex with sulphuric acid 1 g, water 90 ml, sulphuric acid 10 ml). The TLC plates were developed at 80°C for 25 min and then imaged on a digital scanner. The TAG content was quantified by analysing the resulting TIFF images using the densitometry tool in ImageJ software. All reagents were purchased from Sigma-Aldrich.

## Lipid export block

The shuttling of lipids between organs was abolished by downregulating the apolipoprotein lipid transfer protein (LTP) through heat-shock-activated acute RNAi expression based on the pFRiPE system (*Marois and Eaton, 2007*). In the larva, this manipulation prevents the loading of gut-originated medium-chain diacylglycerides, which are a dominant component of circulating lipids, onto the haemolymph carrier Lipophorin (Lpp) and leads to the accumulation of stored lipid in the larval gut in triglyceride form (*Palm et al., 2012*). The downregulation of *LTP* from the fat body driver *lpp-Gal4* was triggered in virgin females by 1 hr heat-shock at 37°C; after 6 hr, the guts were dissected for neutral lipid detection using Oil Red O staining.

## Oil Red O stainings

Fly guts were dissected from flies immobilised on ice and were then fixed in a solution of 4% formaldehyde for 45 min. Guts were washed in consecutive applications of phosphate buffered saline (PBS), double-distilled water, and a 60% isopropanol solution. Oil Red O (Sigma-Aldrich) stock was prepared as a 0.1% solution in isopropanol, then a freshly prepared working solution (a 6:4 dilution in water) was added for 20 min to the guts, then washed in 60% isopropanol and water. The preparations were mounted in glycerol for analysis, and the posterior midgut was imaged using either a Zeiss Axioplan stereo microscope equipped with Nomarski optics or an Olympus BX53 phase contrast microscope equipped with a 4×/0.13 UPlanFLN lens through CellSens software (Olympus, Japan). The resulting TIFF files were analysed quantitatively using a custom ImageJ script: the gut was manually outlined as a ROI using the polygon tool, then the RGB channels were split and the red channel subtracted from the green to eliminate background (grey) signal. The mean intensity of the resulting signal within the ROI was calculated with the built-in Analyse Particles function.

## Statistics and figure preparation

All statistical analyses were carried out in the R environment (*R Development Core Team, 2014*). Comparisons between normally distributed groups were carried out using Student's *t*-test (R function t.test), unpaired, two-tailed and incorporating Welch's correction to account for unequal variances, followed by Bonferroni-Holm correction when multiple comparisons were applied. qPCR data were analysed comparing the housekeeping-subtracted Cts of experimentally matched virgin and mated samples, thus using paired t-test, one-tailed when confirming previous reporter experiments (*Figure 2E,F*), and two-tailed when no prediction could be made (panel H in *Figure 3—figure supplement 1*). Count data with a distinctly non-normal distribution (specifically, pH3 counts) were fitted with a negative binomial model (R function glm.nb from MASS package, *Venables and Ripley, 2002*) followed by likelihood ratio tests (R function anova.negbin from MASS package). Rank-based experiments were analysed with the Mann-Whitney-Wilcoxon rank sum test (R function wilcox.test). All graphs were generated in R using a custom script based on the base boxplot function superimposed with individual data points plotted with the beeswarm function (package beeswarm). Confocal and bright-field images shown in conjunction were always acquired with the same settings as part of a single experiment. For visualisation purposes, level and channel adjustments were applied using Adobe Photoshop CS6 to the confocal images shown in figure panels (the same correction in all comparable images), but all quantitative analyses were carried out on unadjusted raw images or maximum projections. In all figures, * indicates $0.05 > p \geq 0.01$, ** indicates $0.01 > p \geq 0.001$, and *** indicates $p < 0.0001$.

## Acknowledgements

We thank Antoine Ducuing for his assistance with the initial SREBP experiments and Suzanne Eaton and Wilhelm Palm for sharing lipid transport data and reagents prior to publication. We are grateful to Frederic Bernard, Edward Dubrowsky, Suzanne Eaton, Marek Jindra, Christen Mirth, Isabel Palacios, Julien Royet, Alex Shingleton, and Graham Thomas for reagents. We also thank Thomas Carroll for statistical advice, and Dafni Hadjieconomou, Bruno Hudry, Bryn Owen, Esmeralda Parra-Peralbo, Daniel Perea and Marc Tatar for comments on the manuscript. This work was funded by grants from the Wellcome Trust, European Research Council and the Medical Research Council (WT083559, ERCStG 310411 and intramural to IM-A), Generalitat Valenciana (PROMETEO II/2013/001 to MD), Spanish Ministry of Science (BFU2009-09074, SAF2012-35181, CSD2007-00023 to MD), Botin Foundation (to MD), and the Singapore National Research Foundation (RF2010-06 to JYY). TR held a postdoctoral fellowship from the Deutsche Forschungsgemeinschaft and IM-A is a member of, and is supported by, the EMBO Young Investigator Programme.

## Additional information

### Funding

| Funder | Grant reference | Author |
|---|---|---|
| Wellcome Trust | WT083559 | Irene Miguel-Aliaga |
| European Research Council (ERC) | ERCStG 310411 | Irene Miguel-Aliaga |
| Medical Research Council (MRC) | intramural | Irene Miguel-Aliaga |
| Ministerio de Ciencia e Innovacion | BFU2009-09074 | Maria Dominguez |
| Botin Foundation | | Maria Dominguez |
| National Research Foundation-Prime Minister's office, Republic of Singapore (NRF) | RF2010-06 | Joanne Y Yew |
| Generalitat Valenciana (Regional Government of Valencia) | PROMETEO II/2013/001 | Maria Dominguez |
| Ministerio de Ciencia e Innovacion | CSD2007-00023 | Maria Dominguez |
| Ministerio de Ciencia e Innovacion | SAF2012-35181 | Maria Dominguez |

The funders had no role in study design, data collection and interpretation, or the decision to submit the work for publication.

### Author contributions

TR, JJ, Revising the article, Conception and design, Acquisition of data, Analysis and interpretation of data; PC, Writing the article, Conception and design, Acquisition of data, Analysis and interpretation of data; ZA, Conception and design, Acquisition of data; EB, KJT, Acquisition of data, Contributed unpublished essential data or reagents; JYY, Revising the article, Analysis and interpretation of data; MD, Revising the article, Conception and design, Analysis and interpretation of data; IM-A, Writing the article, Conception and design, Analysis and interpretation of data

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
