## [Decision Letter]

Thank you for sending your work entitled “Direct endocrine remodelling of the adult intestine maximises reproductive success in *Drosophila*” for consideration at *eLife*. Your article has been favorably evaluated by Fiona Watt (Senior editor), a Reviewing editor, and two reviewers.

The Reviewing editor and the reviewers discussed their comments before we reached this decision, and the Reviewing editor has assembled the following comments to help you prepare a revised submission.

The reviewers agree that this is an original, interesting and broadly convincing paper that is suitable in principle for publication in *eLife*. However, in addition to a number of minor points, there were a few more significant issues that should be addressed before the paper can be accepted.

Both the Abstract and the main text are so brief that background and experimental details are omitted. Without adding too much extra text, it should be revised so that it is a complete document. The Abstract should contain important keywords for searching.

The results shown in Figure 3, in which depletion of either of the two JH receptors completely blocks ISC mitoses and gut growth, is strange enough that it might be an artefact. If these gene products are both bona fide JH receptors then we might expect them to have overlapping, redundant functions, and that depleting both (not just one) of them would be necessary to fully block JH action. The authors need to do some controls to ensure that these RNAis are not just killing the stem cells, in a JH independent manner; or they need to explain clearly why this is not necessary.

The fecundity assays should also be done with the *esgReDDM* driver to test directly the relevance of gut resizing to fertility. Without some kind of direct test, they will probably have to weaken the argument for this interesting model, which would be a pity.

To understand the corpus allatum ablation experiment, it's important to refer explicitly to the reduction of JH titers that this causes, as described in the [88] paper.

*Reviewer #1 minor comments*:

Title: “Direct endocrine remodeling…”—is it necessary/relevant to claim that this is direct? How can the authors be sure about this?

Title and elsewhere: “maximizes reproductive success”—maximization relative to what? To really assess this, I suppose one would need to measure reproductive success among a large number of wild-type genotypes from natural populations. Is the claim here that reproductive success is maximized/optimized in the wild-type (as measured/defined here) and that any deviation from this state reduces reproductive success? The authors should be careful with their phrasing, especially with regard to any putative adaptive interpretation of their findings. For example, in the Abstract/Summary, they state that the “midgut is dramatically remodelled to maximise reproductive success”, and I am wondering how certain one can be about this rather strong claim. I would say that their results are consistent with such an adaptive interpretation but they do not prove it.

Title and elsewhere: “reproductive success”—again, here I think the authors should be a bit more careful. Many of their results pertain to fecundity rather than fertility. Reproductive success in the strict sense refers to the number of surviving offspring (thus to fertility, not to fecundity). From the Methods section it appears that the authors have indeed also measured fertility but, alas, I cannot find these data: all the plots seem to show fecundity (at least that is what the axes say), not fertility. Also note that, unlike defined in the Methods (in the subsection headed “Fertility and fecundity experiments”), viability (=egg to adult survival, which is a proportion or percentage) is *not* the same as fertility. Fertility is the number of total eggs produced times viability, giving the total number of eggs produced from which adult flies eclose. Moreover, importantly, as far as I can tell, the authors have measured fecundity only over the first 6 days of adulthood. This is hardly an accurate measure of (lifetime) reproductive success! It might very well be that minor early differences in fecundity will vanish with age so that there won't be any net difference between two groups claimed to be different. On the other hand, since pek fecundity is reached within 3–4 days after eclosion, the authors can argue that they have focused on the most important part of the age-dependent fecundity profile. Nonetheless, the authors should be more careful when talking about “reproductive success”.

Throughout manuscript, especially in the introductory parts: The literature on the connection between fat metabolism and reproduction is not very well covered; many key references are missing. See, for example, the review by [30] in Cell Metabolism and references therein. There is lots of evidence, for example, that curtailing or abolishing reproduction increases fat storage (which is not necessarily inconsistent at all with the notion that mating increases fat metabolism), and that lipids are one of the main energetic currencies of trade-offs between reproduction and somatic maintenance.

Results and Discussion: “also occur in many mammals”—rephrase perhaps, as this implies that the previous sentence was not about mammals?

Results and Discussion: “adaptations”—do the authors mean here “adaptation” in the physiological sense of the word or do they mean evolutionary adaptation? It is unclear. The authors must be careful here.

Results and Discussion, end of the first paragraph: Interestingly, both mating (as reported here) and curtailing reproduction (e.g., via gonadectomy) increase lipid levels, so have equivalent effects, and this is of course not necessarily a contradiction. Remarkably, the same appears to be true for plastic changes in organ size: similar to the increased organ size upon mating, the size of organs is sometimes also observed in some sterile fly mutants, for instance, with the fat body and the corpus allatum being hypertrophied and with transplanation of wildtype ovaries into the sterile flies restoring normal organ size (papers by Winifred Doane, cited in [30]). I think that some of these connections should be discussed.

Results and Discussion, second paragraph: The results regarding JH III are very interesting, but what about the potential effects of JH3-bisepoxide and of methylfarnesoate? Potential effects of these other juvenoid compounds produced by the corpus allatum should at least be mentioned as a caveat, i.e. while JH III is clearly involved in the phenomena observed here, the authors cannot exclude effects also of JH3B and MF.

Results and Discussion, same paragraph: Was it not possible to simultaneously downregulate/silence Met and Gce? As far as I understand these two paralogs only function as an operational JH receptor in *Drosophila* as a heterodimer.

Results and Discussion, third paragraph: *ovo*^*D1*^*—*this is an interesting set of results but, as far as I can tell, the authors did not include a fertile wildtype control for *ovo*^*D1*^ in this experiment, so the only statement can be made from this experiment is about non-mating vs mating. A properly controlled experiment should include 4 groups: WT control vs *ovo*^*D1*^, both mated vs unmated.

Results and Discussion, same paragraph: “might accumulate… peripheral fat stores in the absence of the local (ovary) sink”. Two reactions to this statement: (1) we already know that in many animals sterility (gonadectomy, castratation) causes increased fat accumulation or obesity (cf. [30] and references therein), (2) the *ovo*^*D1*^ mutants are, as far as I know, not properly ovary-less, they just have smaller ovaries than wild-type. Thus, with regard to (1) there is indeed published evidence for the authors’ statement. Similarly, the important result in [88], that corpus allatum ablation impedes fat body development, should be cited and discussed. This result is consistent with the authors' findings.

Results and Discussion: (Endcrine regulation of) Organ size plasticity in *Drosophila*—the authors should cite the papers by Alexander Shingleton here.

Results and Discussion, fourth paragraph, especially the last two sentences: this is interesting and well-written but perhaps a bit on the speculative side.

In the figure legend for Figure 1 and elsewhere: what is a negative binomial ANOVA? By definition an ANOVA is general (not a generalized) linear model which assumes a normal distribution of the errors/residuals. Here the authors seem to have used a generalized linear model with a negative binomial error structure, but I cannot be sure. Please clarify.

In general: In some of their experiments, the authors have used TLC and a (semi-quantitative) quantification of TLC to assess peripheral fat. Why did the authors not use one of the well-established and commercially available kit-based TAG assays?

In the subsection headed “Fly strains”: I am worried that apparently none of the experiments has involved fully co-isogenic, backcrossed wild-type controls which are co-isogenic with the *Gal4* and UAS-RNAi lines used. I think this is probably not a big concern for the majority of experiments, but this might play a role for the fecundity assays since this trait is notoriously quantitative and sensitive to genetic background.

In the subsection headed “Fly strains” and following: did the authors verify the efficiency of RNAi silencing via qRT-PCR? The fact that many manipulations did have phenotypic effects suggests that RNAi has worked but this is of course no proof. In general, I think that if one uses RNAi one should demonstrate that it has indeed reduced mRNA transcript (or even better protein levels).

In the subsection headed “Fly husbandry”: methoprene—how was this dosage determined? Is the methoprene used the pure enantiomere or a racemic mixture of s- and r-methoprene? More details on the product from Sigma should be given.

In the subsection headed “Detection of circulating JH hormone by DART-MS”: synthetic JH III—I am not aware that there is synthetic JH III but I may of course be wrong. Is this not simply proper JH III isolated from Manduca sexta?

In the subsection headed “Fertility and fecundity experiments” and following: I failed to find the fertility experimental data—where are these? Note (see my comment above) that viability is *not* the same as fertility. I am also a bit worried about the temperature shift effects on fecundity: this is rather unnatural. Of course I see the reason for doing this but: (1) were there no GeneSwitch-*Gal4* constructs available that could have been used to restrict silencing to the adult stage? (2) were the (wildtype) controls in these experiments also subjected to the temperature shift? Finally, as mentioned above, measuring fecundity over 6 days may not be a good, accurate predictor of (lifetime) reproductive success.

In the subsection headed “Statistics and figure preparation”: I am confused about “assuming unequal deviances”. Do the authors mean “variances”? “Deviances” is *not* the same as variances; deviances have a special statistical meaning which does not make sense in the context of a *t*-test. And “assuming unequal” variances does not make sense either: a *t*-test assumes normality of residuals and equal variances. Nonetheless, there is in fact a *t*-test which does not require the assumption of equal variances, the Welch *t*-test or Welch-ANOVA, but I am not sure the authors have used this.

Figure 4 strikes me as being unnecessary.

*Reviewer #2 minor comments*:

1) The expression patterns of the *Gal4* drivers need to be described somewhat better (preferably in brief in the main text) and supporting citations or results given. This includes *Mex-, Aug21-* and *esgReDDM-Gal4*.

2) In Figures 2 and 3 the authors have not measured SREBP “activity” as it states in the figure, but levels. Please correct this.

---

## [Author Response]

*Both the Abstract and the main text are so brief that background and experimental details are omitted. Without adding too much extra text, it should be revised so that it is a complete document. The Abstract should contain important keywords for searching*.

Both the Abstract and the main text have been expanded. The Abstract contains keywords such as juvenile hormone, Met and Gce receptors, which were missing in the previous version. The revised main text describes additional experiments (detailed below in response to other points), and includes 39 new references, clarifications and discussions requested by the reviewers (also outlined in subsequent points). As a result, the old Results and Discussion content has been subdivided into additional sections (Results and three Discussion sections).

*The results shown in*
Figure 3*, in which depletion of either of the two JH receptors completely blocks ISC mitoses and gut growth, is strange enough that it might be an artefact. If these gene products are both bona fide JH receptors then we might expect them to have overlapping, redundant functions, and that depleting both (not just one) of them would be necessary to fully block JH action. The authors need to do some controls to ensure that these RNAis are not just killing the stem cells, in a JH independent manner; or they need to explain clearly why this is not necessary*.

We have tested this by quantifying the number of intestinal progenitors (as the number of *esgReDDM* GFP-positive cells) upon downregulation of *gce*. No changes in the number of progenitors were observed, ruling out stem cell death as a possible reason for the reduction in mitoses. This is now shown in Figure 3—figure supplement 1, panel D. To further confirm that the health of intestinal progenitors in which *gce* has been downregulated is not compromised, we tested their ability to respond to an unrelated proliferative stimulus by treating guts with the ROS-inducing compound paraquat (e.g. [8] Cell Stem Cell). Both the number of *gce*-downregulated progenitors and their pH3 counts were comparable to those of controls in response to paraquat. These results further indicate that downregulation of JH signalling in adult intestinal progenitors does not compromise their viability or function—at least within the timeframe of our experiments—and are now shown in Figure 3—figure supplement 1 panels E-F.

Gain-of-function experiments further show the specificity of the stem cell JH phenotypes; both methoprene treatment and overexpression of the JH target *Kr-h1* specifically in adult intestinal progenitors lead to increased proliferation/gut size in virgin females (Figure 3, and Figure 3—figure supplement 1 panels I and J respectively).

We have also used a viable *Met*^*27*^ mutant to confirm RNAi phenotypes, and to further confirm that the actions of *Met* and *gce* in the intestine are not redundant. As the box plots in Figure 5 show, the mating-induced increases in proliferation (assessed by pH3 staining) and midgut diameter do not occur in *Met*^*27*^ female flies. However, the lack of *Met* expression in these mutants is constitutive and in all tissues, and, possibly as a result, their basal intestinal proliferation and midgut size seem higher than those of wild-type flies, so we feel that the RNAi experiments described in the manuscript are more rigorously controlled for potential temporal and/or non-cell autonomous effects.

Author response image 1.**DOI:**
http://dx.doi.org/10.7554/eLife.06930.013

Met has been shown to bind both itself and Gce (see [34] Annual Rev Entomol 58: 181–204 for a review). Hence, it is conceivable that the mating-induced increase in ISC proliferation requires (at least transient) Met-Gce heterodimers, in light of the full phenotype resulting from depletion of either one alone. Alternatively, even if Met and Gce carry out equivalent functions independently, they may be haploinsufficient in the intestine; in this situation, a partial reduction in overall Met/Gce function resulting from loss of either gene would render the intestine unable to undergo postmating remodelling.

The intestinal progenitor quantifications, responses to paraquat and the *Kr-h1* experiments are described in the revised Results section: “Juvenile hormone signals directly to adult intestinal progenitors and enterocytes via Met and Gce receptors”. The *Met*^*27*^ experiment and considerations about molecular mechanisms of Met/Gce action are discussed in the revised Discussion under “Hormonal remodelling of adult organs”.

*The fecundity assays should also be done with the* esgReDDM *driver to test directly the relevance of gut resizing to fertility. Without some kind of direct test, they will probably have to weaken the argument for this interesting model, which would be a pity*.

This is a good point. Similar manipulations to those used to assess contributions of enterocytes to egg production can result in reduced egg production when confined to *esg*-positive progenitors (e.g. *esgReDDM > Met -RNAi* or *esgReDDM > gce-RNAi)*. However, during the course of these experiments, we also noticed that, unlike the enterocyte driver *mex-Gal4*, two routinely used *esg-Gal4* intestinal progenitor drivers are also expressed in a few cells in the ovary (see Figure 6, images by Bruno Hudry and Jake Jacobson, Miguel-Aliaga lab). Another routinely used driver, *GS5961*, is also expressed outside the intestine (Nicholson et al (2008) Genetics 178:215) Hence, although our experiments do suggest that intestinal-progenitor-driven resizing also contributes to increased egg production, we do not feel that we can make such a statement in the manuscript, given that the driver may not be exclusively expressed in the intestine – a finding that we mention in this revised version (last Results section). We do, however, disagree that the model is weakened as a result; the enterocyte data clearly shows that at least some of the mating-induced intestinal changes do affect egg production.

Author response image 2.**DOI:**
http://dx.doi.org/10.7554/eLife.06930.014

*To understand the corpus allatum ablation experiment, it's important to refer explicitly to the reduction of JH titers that this causes, as described in the*
[88]
*paper*.

The Yamamoto results are now described in the paper: when the corpus allatum ablation experiment is introduced (in the Results section “Intestinal remodelling is mediated by increased levels of circulating juvenile hormone”), in the first two sections of the Discussion and in the Figure 3 legend.

Reviewer #1 minor comments:

*Title: “Direct endocrine remodeling…”—is it necessary/relevant to claim that this is direct? How* can *the authors be sure about this*?

We conclude that the action of juvenile hormone on the intestine is direct based on the fact that 1) downregulation of the juvenile hormone receptors specifically in adult intestinal progenitors or enterocytes (Figure 3 and Figure 3—figure supplement 1), and 2) activation of the pathway specifically in these cells by *Kr-h1* expression (Figure 3—figure supplement 1) both recapitulate phenotypes resulting from changes in systemic JH signalling and/or or JH analogue feeding. We believe that this is an important finding because it shows that intestinal remodelling is not secondary to the effects of JH on organs such as the ovary or the fat body. We do, however, agree that “direct endocrine remodelling” may not necessarily convey this information, and have revised the Title and Abstract so that “direct” is now absent from the title, but is more extensively explained in the Abstract.

*Title and elsewhere: “maximizes reproductive success”—maximization relative to what? To really assess this, I suppose one would need to measure reproductive success among a large number of wild-type genotypes from natural populations. Is the claim here that reproductive success is maximized/optimized in the wild-type (as measured/defined here) and that any deviation from this state reduces reproductive success? The authors should be careful with their phrasing, especially with regard to any putative adaptive interpretation of their findings. For example, in the Abstract/Summary, they state that the “midgut is dramatically remodelled to maximise reproductive success”, and I am wondering how certain one* can *be about this rather strong claim. I would say that their results are consistent with such an adaptive interpretation but they do not prove it*.

We did not mean to imply success in ecological/evolutionary terms, but acknowledge that the phrasing may be misleading. Therefore, we have amended the relevant sentences and have replaced “success” throughout the manuscript with more neutral/generic (“reproductive output”), or specific/descriptive (e.g. “egg production at the time of peak fertility”) terms throughout the manuscript. We have also included two additional paragraphs (Introduction, second paragraph, and Discussion, first section in the revised manuscript) providing more information regarding the dynamics of egg production in *Drosophila*, and discussing possible long-term effects of intestinal remodelling on reproductive output.

We have also replaced the word “maximise” with “sustain” or “enhance” throughout the manuscript.

*Title and elsewhere: “reproductive success”—again, here I think the authors should be a bit more careful. Many of their results pertain to fecundity rather than fertility. Reproductive success in the strict sense refers to the number of surviving offspring (thus to fertility, not to fecundity). From the Methods section it appears that the authors have indeed also measured fertility but, alas, I cannot find these data: all the plots seem to show fecundity (at least that is what the axes say), not fertility. Also note that, unlike defined in the Methods (in the subsection headed “Fertility and fecundity experiments”), viability (=egg to adult survival, which is a proportion or percentage) is* not *the same as fertility. Fertility is the number of total eggs produced times viability, giving the total number of eggs produced from which adult flies eclose. Moreover, importantly, as far as I* can *tell, the authors have measured fecundity only over the first 6 days of adulthood. This is hardly an accurate measure of (lifetime) reproductive success! It might very well be that minor early differences in fecundity will vanish with age so that there won't be any net difference between two groups claimed to be different. On the other hand, since pek fecundity is reached within 3-4 days after eclosion, the authors* can *argue that they have focused on the most important part of the age-dependent fecundity profile. Nonetheless, the authors should be more careful when talking about “reproductive success”*.

All instances of reproductive success have now been amended as described above. In addition, we now provide additional data showing a specific effect of these genetic manipulations on egg production rather than egg quality/viability (Figure 4—figure supplement 1, panels H and I), and explain these terms whenever they are used in the revised manuscript.

*Throughout manuscript, especially in the introductory parts: The literature on the connection between fat metabolism and reproduction is not very well covered; many key references are missing. See, for example, the review by*
[30]
*in Cell Metabolism and references therein. There is lots of evidence, for example, that curtailing or abolishing reproduction increases fat storage (which is not necessarily inconsistent at all with the notion that mating increases fat metabolism), and that lipids are one of the main energetic currencies of trade-offs between reproduction and somatic maintenance*.

We apologise for this oversight. The links between reproduction, lipid metabolism and ageing are more extensively discussed in the revised manuscript (first and third Discussion sections).

*Results and Discussion, first paragraph: “also occur in many mammals”—rephrase perhaps, as this implies that the previous sentence was not about mammals*?

“Mammals” has been replaced with “animals”.

*Results and Discussion: “adaptations”—do the authors mean here “adaptation” in the physiological sense of the word or do they mean evolutionary adaptation? It is unclear. The authors must be careful here*.

We meant physiological adaptations. We have replaced “possible intestinal adaptations” with “possible intestinal changes occurring during the phase of peak fertility”.

*Results and Discussion, end of the first paragraph: Interestingly, both mating (as reported here) and curtailing reproduction (e.g.,* via *gonadectomy) increase lipid levels, so have equivalent effects, and this is of course not necessarily a contradiction. Remarkably, the same appears to be true for plastic changes in organ size: similar to the increased organ size upon mating, the size of organs is sometimes also observed in some sterile fly mutants, for instance, with the fat body and the corpus allatum being hypertrophied and with transplanation of wildtype ovaries into the sterile flies restoring normal organ size (papers by Winifred Doane, cited in*
[30]*). I think that some of these connections should be discussed*.

Hansen et al. is now mentioned in the revised manuscript. However, we feel that there is a difference between the hypertrophy examples listed by the reviewer (which may passively result, for example, from increased stores in the case of the fat body) and the organ resizing we have reported, which involves an increase in the number of cells and reprogramming of their metabolic state driven by active signals.

*Results and Discussion, second paragraph: The results regarding JH III are very interesting, but what about the potential effects of JH3-bisepoxide and of methylfarnesoate? Potential effects of these other juvenoid compounds produced by the corpus allatum should at least be mentioned as a caveat, i.e. while JH III is clearly involved in the phenomena observed here, the authors cannot exclude effects also of JH3B and MF*.

We looked for signals corresponding to JH3-bisepoxide and methylfarnesoate in the hemolymph extracts and were unable to detect them, suggesting that their concentration is very low. However, the review is right that we cannot formally exclude effects from these two juvenoid compounds and we now mention this explicitly in the revised manuscript (in the Results section “Intestinal remodelling is mediated by increased levels of circulating juvenile hormone X” and in the Materials and methods section “Detection of circulating JH hormone by DART-MS”).

*Results and Discussion, same paragraph: Was it not possible to simultaneously downregulate/silence Met and Gce? As far as I understand these two paralogs only function as an operational JH receptor in Drosophila as a heterodimer*.

We did not see this as a priority because, as discussed earlier, downregulation of either receptor alone fully prevented the mating-triggered increase in proliferation and resizing. Furthermore, although Met can bind itself and Gce, these dimers dissociate in the presence of methoprene or JHIII. Currently available data suggests that Met may mediate its actions by interaction with other nuclear proteins such as Taiman (see [34] Annu Rev Entomol for a comprehensive review). Given the different effects of *Met* and *gce* downregulation on enterocytes and stem cells, these proteins may differ depending on cell type, and their identification is beyond the scope of this manuscript. These considerations are now discussed in the revised manuscript (second Discussion section: “Hormonal remodelling of adult organs”).

*Results and Discussion, third paragraph:* ovo^D1^—*this is an interesting set of results but, as far as I* can *tell, the authors did not include a fertile wildtype control for* ovo^D1^
*in this experiment, so the only statement* can *be made from this experiment is about non-mating versus mating. A properly controlled experiment should include 4 groups: WT control versus* ovo^D1^, *both mated versus unmated*.

We have conducted the experiments in these four groups as requested by the reviewer. The experiments confirm that both the increases in proliferation and midgut diameter and the activation of SREBP occur to a comparable degree in sterile and fertile flies, and are shown in Figure 4—figure supplement 1, panels A to E.

*Results and Discussion, same paragraph: “might accumulate… peripheral fat stores in the absence of the local (ovary) sink”. Two reactions to this statement: (1) we already know that in many animals sterility (gonadectomy, castratation) causes increased fat accumulation or obesity (cf.*
[30]
*and references therein), (2) the* ovo^D1^
*mutants are, as far as I know, not properly ovary-less, they just have smaller ovaries than wild-type. Thus, with regard to (1) there is indeed published evidence for the authors’ statement. Similarly, the important result in*
[88]*, that corpus allatum ablation impedes fat body development, should be cited and discussed. This result is consistent with the authors' findings*.

We thank the reviewer for pointing out these relevant references. Point 1 is now discussed in the last section of the revised Discussion. Point 2 has been addressed by amending the sentence to “we used sterile *ovo*^*D1*^ females to quantify lipid content, reasoning that it might accumulate in gut and/or peripheral fat stores in the absence of the local ovarian sink” in the revised Results section “Mating-triggered intestinal remodelling maximises reproductive output”. We have also mentioned the effect of JH on the fat body reported by Yamamoto et al. (in the second section of the revised Discussion).

*Results and Discussion: (Endcrine regulation of) Organ size plasticity in* Drosophila*—the authors should cite the papers by Alexander Shingleton here*.

The papers by Alex Shingleton are very nice and a couple of them have been cited elsewhere in the manuscript but, to our knowledge, none of them have shown plasticity in adult organs (as in post-eclosion changes in organ size that do not arise during development).

*Results and Discussion, fourth paragraph, especially the last two sentences: this is interesting and well-written but perhaps a bit on the speculative side*.

We agree that it these ideas are speculative and, as such, we introduce them as “intriguing possibilities”. Given that these are the final sentences of the manuscript, we feel that it is appropriate to propose hypotheses and future research directions, and would like to retain them.

*In the figure legend for*
Figure 1
*and elsewhere: what is a negative binomial ANOVA? By definition an ANOVA is general (not a generalized) linear model which assumes a normal distribution of the errors/residuals. Here the authors seem to have used a generalized linear model with a negative binomial error structure, but I cannot be sure. Please clarify*.

We apologise for this mistake. Because pH3 experiments yielded count (discrete) data with a distinctly non-normal distribution, we fitted these data with a negative binomial generalised linear model (R function glm.nb from MASS package), which takes into account the asymmetric nature and overdispersion of count values (Bliss and Fisher (1953) Biometrics 9, 176–200), and then used likelihood ratio tests (R function anova.negbin from MASS package) to obtain p values. We have explained this analysis in Materials and methods (under “Statistics and Figure Preparation”) and have replaced ANOVA with GLM in the relevant figure legends.

*In general: In some of their experiments, the authors have used TLC and a (semi-quantitative) quantification of TLC to assess peripheral fat. Why did the authors not use one of the well-established and commercially available kit-based TAG assays*?

The kit-based TAG assays are subject to experimental error from two sources:

1) Pigmentation from the eyes or cuticle in the extract can contribute to the colorimetric read-out used for kit-based assays.

2) The kits do not strictly measure TAGs alone. The enzymatic reaction used to produce the colour signal also cleaves other molecules with glycerol backbones such as diacylglycerides and monoacylglycerides. Thus, by using a kit, the colorimetric signal would reflect released glycerol from TAGs as well as other glycerides.

TLC used under the conditions described in the manuscript is capable of separating TAGs from other classes of glycero-lipids. Pigmentation also does not contribute to the TAG signal since the pigments do not have the same retention index (i.e., they do not migrate in the same band) as TAGs.

Previous work comparing commercial kits to TLC for TAG quantitation found the latter to be less sensitive and less accurate. For details see Al-Anzi et al. (2010) PLoS One (doi: 10.1371/journal.pone.0012353).

*In the subsection headed “Fly strains”: I am worried that apparently none of the experiments has involved fully co-isogenic, backcrossed wild-type controls which are co-isogenic with the* Gal4 *and UAS-RNAi lines used. I think this is probably not a big concern for the majority of experiments, but this might play a role for the fecundity assays since this trait is notoriously quantitative and sensitive to genetic background*.

We agree that fecundity is very sensitive to genetic background, which is why we avoid working with isogenic genetic backgrounds that might give phenotypes that cannot be replicated in other genetic backgrounds. For this reason, we opted for testing fecundity in different genetic backgrounds with their appropriate controls. Between Figure 4 and Figure 4—figure supplement 1, we tested 7 independent RNAi lines belonging to 3 different collections (VDRC KK, VDRC GD and TRIP, i.e. 3 different genetic backgrounds), and used genetically matched controls for each collection (for example, for the TRiP RNAi lines, *UAS* control flies were genetically matched and contained an equivalent valium10 *UAS* vector containing the *GFP* coding sequence inserted into the same site as the hairpin construct in the TRiP RNAi lines, the attp2 site). This is now explicitly stated in the Materials and methods section under “Fly strains”.

*In the subsection headed “Fly strains” and following: did the authors verify the efficiency of RNAi silencing* via *qRT-PCR? The fact that many manipulations did have phenotypic effects suggests that RNAi has worked but this is of course no proof. In general, I think that if one uses RNAi one should demonstrate that it has indeed reduced mRNA transcript (or even better protein levels)*.

We have verified that the RNAi transgenes KK100638 and KK101814 downregulate *Met* and *gce* transcripts respectively using qPCR, as requested by the reviewer. This data is now shown in Figure 3—figure supplement 1, panel B. We also provide qPCR data showing downregulation of *Kr-h1* (KK107935) and SREBP (GD37640 and TRiP34073). This data is now shown in Figure 3—figure supplement 1, panel K.

*In the subsection headed “Fly husbandry”: methoprene*—*how was this dosage determined? Is the methoprene used the pure enantiomere or a racemic mixture of s- and r-methoprene? More details on the product from Sigma should be given*.

Regarding dosage: this concentration was chosen in a pilot dilution test from 0.5 to 7.5 mM as the one which induced activation of the *SREBP-Gal4* reporter to levels comparable to mating, and corresponds to approximately half of the concentration used in a previous study ([23] Evolution). This is now clearly stated under Fly Husbandry in Materials and methods. Re: the chemical nature of the methoprene used. Although the Sigma product sheet does not specific whether their methoprene is pure or a racemic mixture, we have checked with Sigma, who has confirmed that it is the racemic mixture. These details and the Sigma product number have been provided in the revised Materials and methods under “Fly husbandry”.

*In the subsection headed “Detection of circulating JH hormone by DART-MS”: synthetic JH III—I am not aware that there is synthetic JH III but I may of course be wrong. Is this not simply proper JH III isolated from Manduca sexta*?

No information is given in the product sheet from Santa Cruz Biotechnology so we have replaced “synthetic JHIII” with “a JHIII standard”, and provide details of product number as well as source in the revised Materials and methods under “Detection of circulating JH hormone by DART-MS”.

*In the subsection headed “Fertility and fecundity experiments” and following: I failed to find the fertility experimental data—where are these? Note (see my comment above) that viability is* not *the same as fertility*.

As mentioned earlier, in addition to testing egg production, we have also tested egg viability in a subset of experiments, and have included these data in the revised manuscript in Figure 4—figure supplement 1 (panels H and I). As requested by the reviewer, we have also amended all references to fertility throughout the manuscript, either replacing them with terms such as “progeny viability” in our experiments and in the Materials and methods section, or explaining the concept further in the revised Introduction and Discussion (e.g. “lifetime fertility” or “at the time of peak fertility”).

*I am also a bit worried about the temperature shift effects on fecundity: this is rather unnatural. Of course I see the reason for doing this but: (1) were there no GeneSwitch-*Gal4 *constructs available that could have been used to restrict silencing to the adult stage? (2) were the (wildtype) controls in these experiments also subjected to the temperature shift? Finally, as mentioned above, measuring fecundity over 6 days may not be a good, accurate predictor of (lifetime) reproductive success*.

In our experience, the RU486 ligand used to activate GeneSwitch constructs can also have other effects, independent of the transgene. Depending on the genetic background, RU486 feeding of control flies causes variability in egg laying and lifespan. Indeed, concerns regarding RU effects in mated female flies in particular have been recently reported (Landis et al. (2015) Aging 7(1):53–69), and may well stem from RU interfering with this intestinal remodelling. For these reasons, we favour the use of adult-specific, temperature -inducible downregulations. In response to the reviewer’s question, absolutely, the controls are also subjected to the same temperature shifts. This is now clearly stated in the revised Materials and methods under “Fly husbandry”. Finally, the point regarding reproductive success has also been addressed as described earlier.

*In the subsection headed “Statistics and figure preparation”: I am confused about “assuming unequal deviances”. Do the authors mean “variances”? “Deviances” is* not *the same as variances; deviances have a special statistical meaning which does not make sense in the context of a* t*-test. And “assuming unequal” variances does not make sense either: a* t*-test assumes normality of residuals and equal variances. Nonetheless, there is in fact a* t*-test which does not require the assumption of equal variances, the Welch* t*-test or Welch-ANOVA, but I am not sure the authors have used this*.

We apologise for this mistake. We did indeed mean variances and the test did include Welch’s correction. This has now been amended in the “Statistics and figure preparation” of the revised Materials and methods.

Figure 4
*strikes me as being unnecessary.*

We agree that readers who are familiar with the anatomy of adult *Drosophila* may find this figure panel unnecessary, but it may help non-*Drosophila* researchers interpret the experiments related to the intestinal effects on egg production. For this reason, we would like to retain it.

*Reviewer #2 minor comments*:

*1) The expression patterns of the* Gal4 *drivers need to be described somewhat better (preferably in brief in the main text) and supporting citations or results given. This includes* Mex-, Aug21*- and* esgReDDM-Gal4.

These are now described when first used in the revised Results section, and each mention is followed by the relevant reference.

*2) In*
Figures 2 and 3
*the authors have not measured SREBP “activity” as it states in the figure, but levels. Please correct this*.

The SREBP reporter used in Figures 2 and 3 is unusual in that, rather than reporting gene expression like most *Gal4* drivers, is only turned on in cells in which SREBP is proteolytically processed and therefore active (see [41] Cell metabolism 3, 439–448 for details). So by quantifying GFP levels, as the Y axis of the graph shows (“Ranked GFP levels”), we obtain information about activity. We feel that it is important to highlight that we are visualising activity in the image heading so as to distinguish it from transcript levels, which we have also quantified (e.g. Figure 2). However, we acknowledge that it can be confusing and, for this reason, we are providing a more detailed explanation including the Kunte et al. reference in both the revised Results section and the legend of Figure 2.